# ZO-AdaMM: Zeroth-Order Adaptive Momentum Method for Black-Box Optimization

**Xiangyi Chen**[1,*]  **Sijia Liu**[2,*]  **Kaidi Xu**[3,*]  **Xingguo Li**[4,*]
**Xue Lin**[3]  **Mingyi Hong**[1]  **David Cox**[2]
[1]University of Minnesota, USA
[2]MIT-IBM Watson AI Lab, IBM Research, USA
[3]Northeastern University, USA
[4]Princeton University, USA

## Abstract

The adaptive momentum method (AdaMM), which uses past gradients to update descent directions and learning rates simultaneously, has become one of the most popular first-order optimization methods for solving machine learning problems. However, AdaMM is not suited for solving black-box optimization problems, where explicit gradient forms are difficult or infeasible to obtain. In this paper, we propose a zeroth-order AdaMM (ZO-AdaMM) algorithm, that generalizes AdaMM to the gradient-free regime. We show that the convergence rate of ZO-AdaMM for both convex and nonconvex optimization is roughly a factor of $O(\sqrt{d})$ worse than that of the first-order AdaMM algorithm, where $d$ is problem size. In particular, we provide a deep understanding on why Mahalanobis distance matters in convergence of ZO-AdaMM and other AdaMM-type methods. As a byproduct, our analysis makes the first step toward understanding adaptive learning rate methods for nonconvex constrained optimization. Furthermore, we demonstrate two applications, designing per-image and universal adversarial attacks from black-box neural networks, respectively. We perform extensive experiments on ImageNet and empirically show that ZO-AdaMM converges much faster to a solution of high accuracy compared with 6 state-of-the-art ZO optimization methods.

## 1   Introduction

The development of gradient-free optimization methods has become increasingly important to solve many machine learning problems in which explicit expressions of the gradients are expensive or infeasible to obtain [1–7]. *Zeroth-Order (ZO)* optimization methods, one type of gradient-free optimization methods, mimic first-order (FO) methods but approximate the full gradient (or stochastic gradient) through random gradient estimates, given by the difference of function values at random query points [8, 9]. Compared to Bayesian optimization, derivative-free trust region methods, genetic algorithms and other types of gradient-free methods [10–13], ZO optimization has two main advantages: a) ease of implementation, via slight modification of commonly-used gradient-based algorithms, and b) comparable convergence rates to first-order algorithms.

Due to the stochastic nature of ZO optimization, which arises from both data sampling and random gradient estimation, existing ZO methods suffer from large variance of the noisy gradient compared to FO stochastic methods [14]. In practice, this causes poor convergence performance and/or function query efficiency. To partially mitigate these issues, ZO sign-based SGD (ZO-signSGD) was proposed by [14] with the rationale that taking the sign of random gradient estimates (i.e., normalizing gradient estimates elementwise) as the descent direction improves the robustness of gradient estimators

---

to stochastic noise. Although ZO-signSGD has faster convergence speed than many existing ZO algorithms, it is only guaranteed to converge to a neighborhood of a solution. In the FO setting, taking the sign of a stochastic gradient as the descent direction gives rise to signSGD [15]. The use of sign of stochastic gradients also appears in adaptive momentum methods (AdaMM) such as Adam [16], RMSProp [17], AMSGrad [18], Padam [19], and AdaFom [20]. Indeed, it has been suggested by [21] that AdaMM enjoy dual advantages of sign descent and variance adaption.

Considering the motivation of ZO-signSGD and the success of AdaMM in FO optimization, one question arises: Can we generalize AdaMM to the ZO regime? To answer this question, we develop the zeroth-order adaptive momentum method (ZO-AdaMM) and analyze its convergence properties in both convex and nonconvex settings for constrained optimization.

**Contributions** *Theoretically*, for both convex and nonconvex optimization, we show that ZO-AdaMM is roughly a factor of $O(\sqrt{d})$ worse than that of the FO AdaMM algorithm, where $d$ is the number of optimization variables. We also show that the *Euclidean* projection based AdaMM-type methods could suffer non-convergence issues for constrained optimization. This highlights the necessity of *Mahalanobis distance* based projection. And we establish the Mahalanobis distance based convergence analysis, which makes the first step toward understanding adaptive learning rate methods for nonconvex constrained optimization.

*Practically*, we formalize the experimental comparison of ZO-AdaMM with 6 state-of-the-art ZO algorithms in the application of black-box adversarial attacks to generate both per-image and universal adversarial perturbations. Our proposal could provide an experimental benchmark for future studies on ZO optimization. Code to reproduce experiments is released at the link `https://github.com/KaidiXu/ZO-AdaMM`.

**Related work** Many types of ZO algorithms have been developed, and their convergence rates have been rigorously studied under different problem settings. We highlight some recent works as below. For unconstrained stochastic optimization, ZO stochastic gradient descent (ZO-SGD) [9] and ZO stochastic coordinate descent (ZO-SCD) [22] were proposed, which have $O(\sqrt{d}/\sqrt{T})$ convergence rate, where $T$ is the number of iterations. Compared to FO stochastic algorithms, ZO optimization suffers a slowdown dependent on the variable dimension $d$, e.g., $O(\sqrt{d})$ for ZO-SGD and ZO-SCD. In [23], the tightness of the dimension-dependent factor $O(\sqrt{d})$ has been proved in the framework of ZO stochastic mirror descent (ZO-SMD). In order to further improve the iteration complexity of ZO algorithms, the technique of variance reduction was applied to ZO-SGD and ZO-SCD, leading to ZO stochastic variance reduced algorithms with an improved convergence rate in $T$, namely, $O(d/T)$ [24–26]. This improvement is aligned with ZO gradient descent (ZO-GD) for deterministic nonconvex programming [8]. Moreover, ZO versions of proximal SGD (ProxSGD) [27], Frank-Wolfe (FW) [28, 2, 29], and online alternating direction method of multipliers (OADMM) [1, 30] have been developed for constrained optimization. Aside from the recent works on ZO algorithms mentioned before, there is rich literature in derivative-free optimization (DFO). Traditional DFO methods can be classified into direct search-based methods and model-based methods. Both the two types of methods are mostly iterative methods. The difference is that direct search-based methods refine their search directions based on the queried function values directly, while a model-based method builds a model that approximates the function to be optimized and updates the search direction based on the model. Representative methods developed in DFO literature include NOMAD [31, 32], PSWarm [33], Cobyla [34], and BOBYQA [35]. More comprehensive discussions on DFO methods can be found in [36, 37].

## 2 Preliminaries: Gradient Estimation via ZO Oracle

The ZO gradient estimate of a function $f$ is constructed by the forward difference of two function values at a random unit direction:

$$\hat{\nabla} f(\mathbf{x}) = (d/\mu)[f(\mathbf{x} + \mu\mathbf{u}) - f(\mathbf{x})]\mathbf{u}, \tag{1}$$

where $\mathbf{u}$ is a random vector drawn uniformly from the sphere of a unit ball, and $\mu > 0$ is a small step size, known as the smoothing parameter. In many existing work such as [8, 9], the random direction vector $\mathbf{u}$ was drawn from the standard Gaussian distribution. Here the use of uniform distribution ensures that the ZO gradient estimate (1) is defined in a bounded space rather than the whole real space required for Gaussian. As will be evident later, the boundedness of random gradient estimates is one of important conditions in the convergence analysis of ZO-AdaMM.

The rationale behind the ZO gradient estimate (1) is that although it is a biased approximation to the true gradient of $f$, it is *unbiased* to the gradient of the randomized smoothing version of $f$ with parameter $\mu$ [23, 24, 30], i.e.,

$$f_\mu(\mathbf{x}) = \mathbb{E}_{\mathbf{u} \sim U_\mathrm{B}}[f(\mathbf{x} + \mu\mathbf{u})], \tag{2}$$

where $\mathbf{u} \sim U_\mathrm{B}$ denotes the uniform distribution over the unit Euclidean ball B. We review properties of the smoothing function (2) and connections to the ZO gradient estimator (1) in Appendix 1.

## 3  AdaMM from First to Zeroth Order

Consider a stochastic optimization problem of the generic form

$$\min_{\mathbf{x} \in \mathcal{X}} f(\mathbf{x}) = \mathbb{E}_{\boldsymbol{\xi}}[f(\mathbf{x}; \boldsymbol{\xi})], \tag{3}$$

where $\mathbf{x} \in \mathbb{R}^d$ are optimization variables, $\mathcal{X}$ is a closed convex set, $f$ is a differentiable (possibly nonconvex) objective function, and $\boldsymbol{\xi}$ is a certain random variable that captures environmental uncertainties. In problem (3), if $\boldsymbol{\xi}$ obeys a uniform distribution built on empirical samples $\{\boldsymbol{\xi}_i\}_{i=1}^n$, then we recover a finite-sum formulation with the objective function $f(\mathbf{x}) = \frac{1}{n}\sum_{i=1}^n f(\mathbf{x}; \boldsymbol{\xi}_i)$.

**First-order AdaMM in terms of AMSGrad [18].** We specify the algorithmic framework of AdaMM by AMSGrad [18], a modified version of Adam [16] with convergence guarantees for both convex and nonconvex optimization. In the algorithm, the descent direction $\mathbf{m}_t$ is given by an exponential moving average of the past gradients. The learning rate $r_t$ is adaptively penalized by a square root of exponential moving averages of squared past gradients. It has been proved in [18, 20, 38, 39] that AdaMM can reach $O(1/\sqrt{T})^2$ convergence rate. Here we omit its possible dependency on $d$ for simplicity, but more accurate analysis will be provided later in Section 4 and 5.

**ZO-AdaMM.** By integrating AdaMM with the random gradient estimator (1), we obtain ZO-AdaMM in Algorithm 1. Here the square root, the square, the maximum, and the division operators are taken elementwise. Also, $\Pi_{\mathcal{X},\mathbf{H}}(\mathbf{a})$ denotes the projection operation under Mahalanobis distance with respect to $\mathbf{H}$, i.e., $\arg\min_{\mathbf{x}\in\mathcal{X}} \|\sqrt{\mathbf{H}}(\mathbf{x} - \mathbf{a})\|_2^2$. If $\mathcal{X} = \mathbb{R}^d$, the projection step simplifies to $\mathbf{x}_{t+1} = \mathbf{x}_t - \alpha_t \hat{\mathbf{V}}_t^{-1/2}\mathbf{m}_t$. Clearly, $\alpha_t \hat{\mathbf{V}}_t^{-1/2}$ and $\mathbf{m}_t$ can be interpreted as the adaptive learning rate and the momentum-type descent direction, which adopt exponential moving averages as follows,

---

**Algorithm 1** ZO-AdaMM

**Input:** $\mathbf{x}_1 \in \mathcal{X}$, step sizes $\{\alpha_t\}_{t=1}^T$, $\beta_{1,t}, \beta_2 \in (0, 1]$, and set $\mathbf{m}_0$, $\mathbf{v}_0$ and $\hat{\mathbf{v}}_0$
**for** $t = 1, 2, \ldots, T$ **do**
    let $\hat{\mathbf{g}}_t = \hat{\nabla} f_t(\mathbf{x}_t)$ by (1), $f_t(\mathbf{x}_t) := f(\mathbf{x}_t; \boldsymbol{\xi}_t)$
    $\mathbf{m}_t = \beta_{1,t}\mathbf{m}_{t-1} + (1 - \beta_{1,t})\hat{\mathbf{g}}_t$
    $\mathbf{v}_t = \beta_2\mathbf{v}_{t-1} + (1 - \beta_2)\hat{\mathbf{g}}_t^2$
    $\hat{\mathbf{v}}_t = \max(\hat{\mathbf{v}}_{t-1}, \mathbf{v}_t)$, and $\hat{\mathbf{V}}_t = \mathrm{diag}(\hat{\mathbf{v}}_t)$
    $\mathbf{x}_{t+1} = \Pi_{\mathcal{X}, \sqrt{\hat{\mathbf{V}}_t}}(\mathbf{x}_t - \alpha_t \hat{\mathbf{V}}_t^{-1/2}\mathbf{m}_t)$
**end for**

---

$$\mathbf{m}_t = \sum_{j=1}^t \left[ \left(\prod_{k=1}^{t-j}\beta_{1,t-k+1}\right)(1 - \beta_{1,j})\hat{\mathbf{g}}_j \right], \quad \mathbf{v}_t = (1 - \beta_2)\sum_{j=1}^t (\beta_2^{t-j}\hat{\mathbf{g}}_j^2). \tag{4}$$

Here we assume that $\mathbf{m}_0 = \mathbf{0}$, $\mathbf{v}_0 = \mathbf{0}$ and $0^0 = 1$ by convention, and let $\hat{\mathbf{g}}_t = \hat{\nabla}f_t(\mathbf{x}_t)$ by (1) with $f_t(\mathbf{x}_t) := f(\mathbf{x}_t; \boldsymbol{\xi}_t)$.

**Motivation and rationale behind ZO-AdaMM.** First, gradient normalization helps noise reduction in ZO optimization as shown by [6, 14]. In the similar spirit, ZO-AdaMM also normalizes the descent direction $\mathbf{m}_t$ by $\sqrt{\hat{\mathbf{v}}_t}$. Particularly, compared to AdaMM, ZO-AdaMM prefers a small value of $\beta_2$ in practice, implying a strong favor to normalize the current gradient estimate; see Fig A1 in Appendix. In the extreme case of $\beta_{1,t} = \beta_2 \to 0$ and $\hat{\mathbf{v}}_t = \mathbf{v}_t$, ZO-AdaMM could reduce to ZO-signSGD [14] since $\hat{\mathbf{V}}_t^{-1/2}\mathbf{m}_t = \mathbf{m}_t/\sqrt{\mathbf{v}_t} = \hat{\mathbf{g}}_t/\sqrt{\hat{\mathbf{g}}_t^2} = \mathrm{sign}(\hat{\mathbf{g}}_t)$ known from (4). However, the downside of ZO-signSGD is its worse convergence accuracy than ZO-SGD, i.e., it only converges to a neighborhood of a stationary point even for unconstrained optimization. Compared to ZO-signSGD, ZO-AdaMM is able to cover ZO-SGD as a special case when $\beta_{1,t} = 0$, $\beta_2 = 1$, $\mathbf{v}_0 = \mathbf{1}$ and $\hat{\mathbf{v}}_0 \le \mathbf{1}$ from Algorithm 1. Thus, we hope that with appropriate choices of $\beta_{1,t}$ and $\beta_2$, ZO-AdaMM could enjoy dual advantages of ZO-signSGD and ZO-SGD. Another motivation comes from the possible presence of time-dependent gradient priors [40]. Given this, the use of past gradients in momentum also helps noise reduction.

**Why is ZO-AdaMM difficult to analyze?** The convergence analysis of ZO-AdaMM becomes significantly more challenging than existing ZO methods due to the involved coupling among stochastic sampling, ZO gradinet estimation, momentum, adaptive learning rate, and projection operation. In particular, the use of Mahalanobis distance in projection step plays a key role on convergence guarantees. And the conventional variance bound on ZO gradient estimates is insufficient to analyze the convergence of ZO-AdaMM due to the use of adaptive learning rate. In the next sections, we will carefully study the convergence of ZO-AdaMM under different settings.

## 4  Convergence Analysis of ZO-AdaMM for Nonconvex Optimization

In this section, we begin by providing a deep understanding on the importance of Mahalanobis distance used in ZO-AdaMM (Algorithm 1), and then introduce the Mahalanobis distance based convergence analysis for both unconstrained and constrained nonconvex optimization. Our analysis makes the first step toward understanding adaptive learning rate methods for nonconvex constrained optimization. Throughout the section, we make the following assumptions.

**A1**: $f_t(\cdot) := f(\cdot; \boldsymbol{\xi}_t)$ has $L_g$-Lipschitz continuous gradient, where $L_g > 0$.

**A2**: $f_t$ has $\eta$-bounded stochastic gradient $\|\nabla f_t(\mathbf{x})\|_\infty \leq \eta$.

### 4.1  Importance of Mahalanobis distance based projection operation

Recall from Algorithm 1 that ZO-AdaMM takes the projection operation $\Pi_{\mathcal{X}, \sqrt{\hat{\mathbf{V}}_t}}(\cdot)$ onto the constraint set $\mathcal{X}$ under Mahalanobis distance with respect to (w.r.t.) $\hat{\mathbf{V}}_t$. In some recent adversarial learning algorithms [41, 42], the Euclidean projection $\Pi_{\mathcal{X}}(\cdot)$ was used in both FO and ZO AdaMM-type methods rather than the Mahalanobis distance based projection in Algorithm 1. However, such an implementation could lead to *non-convergence*: Proposition 1 shows the non-convergence issue of Algorithm 1 using the Euclidean projection operation when solving a simple linear program subject to $\ell_1$-norm constraint. This is an important point which is ignored in design of many algorithms on adversarial training [43].

**Proposition 1** *Consider the following problem*

$$\underset{\mathbf{x}=[x_1, x_2]^T}{\text{minimize}} \; -2x_1 - x_2; \quad \text{subject to } |x_1 + x_2| \leq 1, \tag{5}$$

*then Algorithm 1, initialized by* $\mathbf{x} = [0.5, 0.5]^T$, *using the Euclidean projection* $\Pi_{\mathcal{X}}(\cdot)$ *converges to a fixed point* $[0.5, 0.5]^T$ *rather than a stationary point of* (5).

***Proof***: *The proof investigates a special case of Algorithm 1, projected signSGD; See Appendix 2.1.*

Proposition 1 indicates that replacing the Mahalanobis distance based projection in Algorithm 1 with Euclidean projection will lead to a divergent algorithm, highlighting the importance of using Mahalanobis distance. However, the use of Mahalanobis distance based projection complicates the convergence analysis, especially in constrained optimization. Accordingly, we define a Mahalanobis based convergence measure that can simplify the analysis and can be converted into the traditional convergence measure.

Let $\mathbf{x}^+ = \mathbf{x}_{t+1}$, $\mathbf{x}^- = \mathbf{x}_t$, $\mathbf{g} = \mathbf{m}_t$, $\omega = \alpha_t$ and $\mathbf{H} = \hat{\mathbf{V}}_t^{1/2}$, the projection step of Algorithm 1 can be written in the generic form

$$\mathbf{x}^+ = \underset{\mathbf{x} \in \mathcal{X}}{\arg\min}\{\langle \mathbf{g}, \mathbf{x} \rangle + (1/\omega) D_{\mathbf{H}}(\mathbf{x}, \mathbf{x}^-)\}, \tag{6}$$

where $D_{\mathbf{H}}(\mathbf{x}, \mathbf{x}^-) = \|\mathbf{H}^{1/2}(\mathbf{x} - \mathbf{x}^-)\|^2/2$ gives the Mahalanobis distance w.r.t. $\mathbf{H}$, and $\|\cdot\|$ denotes $\ell_2$ norm. Based on (6), the concept of *gradient mapping* [27] is given by

$$P_{\mathcal{X}, \mathbf{H}}(\mathbf{x}^-, \mathbf{g}, \omega) := (\mathbf{x}^- - \mathbf{x}^+)/\omega. \tag{7}$$

The gradient mapping $P_{\mathcal{X}, \mathbf{H}}(\mathbf{x}^-, \mathbf{g}, \omega)$ yields a natural interpretation: a projected version of $\mathbf{g}$ at the point $\mathbf{x}^-$ given the learning rate $\omega$, yielding $\mathbf{x}^+ = \mathbf{x}^- - \omega P_{\mathcal{X}, \mathbf{H}}(\mathbf{x}^{-1}, \mathbf{g}, \omega)$. We note that different from [27, 44], the gradient mapping in (7) is defined on the projection under the Mahalanobis distance $D_{\mathbf{H}}(\cdot, \cdot)$ rather than the Euclidean distance.

With the aid of (7), we propose the Mahalanobis distance based convergence measure for ZO-AdaMM:

$$\|\mathcal{G}(\mathbf{x}_t)\|^2 := \|\hat{\mathbf{V}}_t^{1/4} P_{\mathcal{X}, \hat{\mathbf{V}}_t^{1/2}}(\mathbf{x}_t, \nabla f(\mathbf{x}_t), \alpha_t)\|^2. \tag{8}$$

If $\mathcal{X} = \mathbb{R}^d$, then the convergence measure (8) reduces to

$$\|\hat{\mathbf{V}}_t^{-1/4} \nabla f(\mathbf{x}_t)\|^2, \tag{9}$$

which corresponds to the squared Euclidean norm of gradient in a linearly transformed coordinate system $\mathbf{y}_t = \hat{\mathbf{V}}_t^{1/4} \mathbf{x}_t$. As will be evident later, the measure (9) can be transformed to the conventional measure $\|\nabla f(\mathbf{x}_t)\|^2$ for unconstrained optimization.

We remark that Mahalanobis (M-) distance facilitates our convergence analysis in an equivalently transformed space, over which the analysis can be generalized from the conventional projected gradient descent framework. To get intuition, let us consider a simpler first-order case with the $\mathbf{x}$-descent step given by Algorithm 1 as $\beta_{1,t} = 0$ and $\mathcal{X} = \mathbb{R}^d$: $\mathbf{x}_{t+1} = \mathbf{x}_t - \alpha \hat{\mathbf{V}}_t^{-1/2} \nabla f(\mathbf{x}_t)$. Note that the ZO case is more involved but follows the same intuition. Upon defining $\mathbf{y}_t \triangleq \hat{\mathbf{V}}_t^{1/4} \mathbf{x}_t$, the $\mathbf{x}$-update can then be rewritten as the update rule in $\mathbf{y}$: $\mathbf{y}_{t+1} = \mathbf{y}_t - \alpha \hat{\mathbf{V}}_t^{-1/4} \nabla f(\mathbf{x}_t)$. Since $\nabla_{\mathbf{y}_t} f(\mathbf{x}_t) = (\frac{\partial \mathbf{x}_t}{\partial \mathbf{y}_t})^T \nabla f(\mathbf{x}_t) = \hat{\mathbf{V}}_t^{-1/4} \nabla f(\mathbf{x}_t)$, the $\mathbf{y}$-update, $\mathbf{y}_{t+1} = \mathbf{y}_t - \alpha \nabla_{\mathbf{y}} f(\mathbf{x}_t)$, obeys the gradient descent framework. In the constrained case, a similar but more involved analysis can be made, showing that the *M-projection in the $\mathbf{x}$-coordinate system* is *equivalent* to the *Euclidean projection in the $\mathbf{y}$-coordinate system* which makes projected gradient descent applicable to the update in $\mathbf{y}$. By contrast, the direct use of *Euclidean projection in the $\mathbf{x}$-coordinate system* leads to *divergence* in ZO-AdaMM (Proposition 1).

## 4.2 Unconstrained nonconvex optimization

We next demonstrate the convergence analysis of ZO-AdaMM for unconstrained nonconvex optimization. In Proposition 2, we begin by exploring the relationship between the convergence measure (9) and ZO gradient estimates; *See Appendix 2.2 for proof.*

**Proposition 2** *Suppose that **A1-A2** hold and let $\mathcal{X} = \mathbb{R}^d$, $\hat{\mathbf{v}}_0^{1/2} \geq c\mathbf{1}$, $f_\mu(\mathbf{x}_1) - \min_{\mathbf{x}} f_\mu(\mathbf{x}) \leq D_f$, $\beta_{1,t} = \beta_1$, $\gamma := \beta_1/\beta_2 < 1$, $\mu = 1/\sqrt{Td}$, and $\alpha_t = 1/\sqrt{Td}$ in Algorithm 1, then ZO-AdaMM yields*

$$\mathbb{E}\left[\left\|\hat{\mathbf{V}}_R^{-1/4} \nabla f(\mathbf{x}_R)\right\|^2\right] \leq \frac{L_g^2}{2c} \frac{d}{T} + 2D_f \frac{\sqrt{d}}{\sqrt{T}} + \frac{L_g(4 + 5\beta_1^2)(1 - \beta_1)}{2(1 - \beta_1)^2(1 - \beta_2)(1 - \gamma)} \frac{\sqrt{d}}{\sqrt{T}}$$

$$+ \frac{2}{c} \mathbb{E}\left[2\eta^2 + \frac{\eta \max_{t \in [T]}\{\|\hat{\mathbf{g}}_t\|_\infty\}}{1 - \beta_1}\right] \frac{d}{T}, \tag{10}$$

*where $\mathbf{x}_R$ is picked uniformly randomly from $\{\mathbf{x}_t\}_{t=1}^T$, and $\hat{\mathbf{g}}_t = \hat{\nabla} f_t(\mathbf{x}_t)$ by (1).*

Proposition 2 implies that the convergence rate of ZO-AdaMM has a dependency on ZO gradient estimates in terms of $G_{\text{zo}} := \max_{t \in [T]}\{\|\hat{\mathbf{g}}_t\|_\infty\}$. Moreover, if we consider the FO AdaMM [20, 38] in which the ZO gradient estimate $\hat{\mathbf{g}}_t$ is replaced with the stochastic gradient, then one can simply assume $\max_{t \in [T]}\{\|\mathbf{g}_t\|_\infty\}$ to be a dimension-independent constant under **A2**. However, in the ZO setting, $G_{\text{zo}}$ is no longer independent of $d$. For example, it could be directly bounded by $\|\hat{\nabla} f(\mathbf{x})\|_2 \leq (d/\mu)\|f(\mathbf{x} + \mu\mathbf{u}) - f(\mathbf{x})\|_2 \leq dL_c$ under the following assumption:

**A3**: $f_t$ is $L_c$-Lipschitz continuous.

In Proposition 3, we show that the dimension-dependency of $G_{\text{zo}}$ can be further improved by using sphere concentration results; *See Appendix 2.3 for proof.*

**Proposition 3** *Under **A3**, $\max\{d, T\} \geq 3$, and given $\delta \in (0, 1)$, then with probability at least $1 - \delta$,*

$$\max_{t \in [T]}\{\|\hat{\mathbf{g}}_t\|_\infty\} \leq 2L_c \sqrt{d \log(dT/\delta)}. \tag{11}$$

Here we provide some insights on Proposition 3. Since the unit random vector used to define $\hat{\mathbf{g}}_t$ is uniformly sampled on a sphere, $\|\hat{\mathbf{g}}_t\|_\infty$ can be improved to $O(\sqrt{d})$ with high probability. This is a tight bound since when the function difference is a constant, the lower bound satisfies $\|\hat{\mathbf{g}}_t\|_\infty = \Omega(\sqrt{d})$ by sphere concentration. It is also not surprising that our bound (11) grows with $T$

since we bound the maximum $\|\hat{\mathbf{g}}_t\|_\infty$ over $T$ realizations with high probability. The time-dependence is required to compensate the growth of the probability that there exists an estimate with the extreme $\ell_\infty$ value versus time. Note that as long as $T$ has polynomial rather than exponential dependency on $d$, we then always have $\max_{t\in[T]}\{\|\hat{\mathbf{g}}_t\|_\infty\} = O(\sqrt{d\log(d)})$. Based on Proposition 2 and Proposition 3, the convergence rate of ZO-AdaMM is provided by Theorem 1; *See Appendix 2.4 for proof.*

**Theorem 1** *Suppose that **A1** and **A3** hold. Given parameter settings in Proposition 2 and 3, then with probability at least $1 - 1/(T\sqrt{d})$, ZO-AdaMM yields*

$$\mathbb{E}\left[\left\|\hat{\mathbf{V}}_R^{-1/4}\nabla f(\mathbf{x}_R)\right\|^2\right] = O\left(\sqrt{d}/\sqrt{T} + d^{1.5}/T\right). \tag{12}$$

We can also extend the convergence rate of ZO-AdaMM in Theorem 1 using the measure $\mathbb{E}[\|\nabla f(\mathbf{x}_R)\|^2]$. Since $\hat{V}_{t,ii}^{-1/2} \geq 1/\max_{t\in[T]}\{\|\hat{\mathbf{g}}_t\|_\infty\}$ (by the update rule), we obtain from (11) that

$$\mathbb{E}\left[\|\nabla f(\mathbf{x}_R)\|^2\right] \leq 2L_c\sqrt{d\log(dT/\delta)}\mathbb{E}\left[\left\|\hat{\mathbf{V}}_R^{-1/4}\nabla f(\mathbf{x}_R)\right\|^2\right]. \tag{13}$$

Theorem 1, together with (13), implies $O(d/\sqrt{T} + d^2/T)$ convergence rate of ZO-AdaMM under the conventional measure. We remark that compared to the FO rate $O(\sqrt{d}/\sqrt{T} + d/T)$ [38] of AdaMM for unconstrained nonconvex optimization under **A1**-**A2**, ZO-AdaMM suffers $O(\sqrt{d})$ and $O(d)$ slowdown on the rate term $O(1/\sqrt{T})$ and $O(1/T)$, respectively. This dimension-dependent slowdown is similar to ZO-SGD versus SGD shown by [9]. We also remark that compared to FO-AdaMM, ZO-AdaMM requires additional **A3** to bound the $\ell_\infty$ norm of ZO gradient estimates.

## 4.3 Constrained nonconvex optimization

To analyze ZO-AdaMM in a general constrained case, one needs to handle the coupling effects from all three factors: momentum, adaptive learning rate, and projection operation. Here we focus on addressing the coupling issue in the last two factors, which yields our results on ZO-AdaMM at $\beta_{1,t} = 0$. This is equivalent to the ZO version of RMSProp [17] with Reddi's convergence fix in [18]. When the momentum factor comes into play, the scenario becomes much more complicated. We leave the answer to the general case $\beta_{1,t} \neq 0$ for future research. Even for SGD with momentum, we are not aware of any successful convergence analysis for stochastic constrained nonconvex optimization.

It is known from SGD [27] that the presence of projection induces a stochastic bias (independent of iteration number $T$) for constrained nonconvex optimization. In Theorem 2, we show that the same challenge holds for ZO-AdaMM. Thus, one has to adopt the variance reduced gradient estimator, which induces higher querying complexity than the estimator (1); *See Appendix 2.5 for proof.*

**Theorem 2** *Suppose that **A1**-**A2** hold, $\hat{\mathbf{v}}_0^{1/2} \geq c\mathbf{1}$, $f_\mu(\mathbf{x}_1) - \min_{\mathbf{x}} f_\mu(\mathbf{x}) \leq D_f$, $\alpha_t = \alpha \leq \frac{c}{L_g}$, $\mu = \frac{1}{\sqrt{Td}}$, and $\beta_{1,t} = 0$ in Algorithm 1, then the convergence rate of ZO-AdaMM under (8) satisfies*

$$\mathbb{E}[\|\mathcal{G}(\mathbf{x}_R)\|^2] \leq \frac{6D_f}{\alpha T} + \frac{3L_g^2 d}{4cT} + \frac{6\eta^2}{c^4 T}(\max_{t\in[T]}\mathbb{E}[\|\hat{\mathbf{g}}_t - f_\mu(\mathbf{x}_t)\|^2] + d\eta^2) + \frac{3c+9}{c}\max_{t\in[T]}\mathbb{E}[\|\hat{\mathbf{g}}_t - f_\mu(\mathbf{x}_t)\|^2],$$

*where $\mathbf{x}_R$ is picked uniformly randomly from $\{\mathbf{x}_t\}_{t=1}^T$, $\mathcal{G}(\mathbf{x})$ has been defined in (8), and $f_\mu$ is the smoothing function of $f$ defined in (2).*

Theorem 2 implies that regardless of the number of iterations $T$, ZO-AdaMM only converges to a solution's neighborhood whose size is determined by the variance of ZO gradient estimates $\max_{t\in[T]}\mathbb{E}[\|\hat{\mathbf{g}}_t - f_\mu(\mathbf{x}_t)\|^2]$. To make this term diminishing, we consider the following variance reduced gradient estimator built on multiple stochastic samples and random direction vectors [14],

$$\hat{\mathbf{g}}_t = \frac{1}{bq}\sum_{j\in\mathcal{I}_t}\sum_{i=1}^q \hat{\nabla}f(\mathbf{x}_t; \mathbf{u}_{i,t}, \boldsymbol{\xi}_j), \quad \hat{\nabla}f(\mathbf{x}_t; \mathbf{u}_{i,t}, \boldsymbol{\xi}_j) := \frac{d[f(\mathbf{x}_t + \mu\mathbf{u}_{i,t}; \boldsymbol{\xi}_j) - f(\mathbf{x}_t; \boldsymbol{\xi}_j)]}{\mu}\mathbf{u}_{i,t}, \tag{14}$$

where $\mathcal{I}_t$ is a mini-batch containing $b$ stochastic samples at time $t$, and $\{\mathbf{u}_{i,t}\}_{i=1}^q$ are $q$ random direction vectors at time $t$. We present the variance of (14) in Lemma 1, whose proof is induced from [14, Proposition 2] by using $\|\nabla f_t\|_2^2 \leq d\|\nabla f_t\|_\infty^2 = d\eta^2$ in **A2**.

**Lemma 1** *Suppose that A1-A2 hold, then for $\mu \leq 1/\sqrt{d}$, the variance of* (14) *yields*

$$\mathbb{E}\left[\|\hat{\mathbf{g}}_t - \nabla f_\mu(\mathbf{x}_t)\|_2^2\right] = O\left(d/b + d^2/q\right). \tag{15}$$

Based on Lemma 1, the rate of ZO-AdaMM in Theorem 2 becomes $\mathbb{E}[\|\mathcal{G}(\mathbf{x}_R)\|^2] = O(d/T + d/b + d^2/q)$. Note that if **A3** holds, then the dimension-dependency can be improved by $O(d)$ factor based on Lemma 1. To the best of our knowledge, even in the FO case we are not aware of existing convergence rate analysis on adaptive learning rate methods for nonconvex contrained optimization.

## 5   Extended Analysis of ZO-AdaMM

**ZO-AdaMM for constrained convex optimization**   Different from the nonconvex case, the convergence of ZO-AdaMM for convex optimization is commonly measured by the average regret $R_T = \mathbb{E}\left[\frac{1}{T}\sum_{t=1}^{T} f_t(\mathbf{x}_t) - \frac{1}{T}\sum_{t=1}^{T} f_t(\mathbf{x}^*)\right]$ [18, 19], where recall that $f_t(\mathbf{x}_t) = f(\mathbf{x}_t; \boldsymbol{\xi}_t)$, and $\mathbf{x}^*$ is the optimal solution. We provide the average regret with the ZO gradient estimates by leveraging its connection to the smoothing function of $f_t$ in Proposition 4; *see Appendix 3.1 for proof.*

**Proposition 4** *Suppose that $\alpha_t = \alpha/\sqrt{t}$, $\beta_{1,t} = \beta_1/t$ with $\beta_{1,1} = \beta_1$, $\beta_1, \beta_2 \in [0,1)$, $\gamma := \beta_1/\sqrt{\beta_2} < 1$ and $\mathcal{X}$ has bounded diameter $D_\infty$, then ZO-AdaMM for convex optimization yields*

$$R_{T,\mu} := \mathbb{E}\left[\frac{1}{T}\sum_{t=1}^{T} f_{t,\mu}(\mathbf{x}_t) - \frac{1}{T}\sum_{t=1}^{T} f_{t,\mu}(\mathbf{x}^*)\right]$$

$$\leq \frac{D_\infty^2 \sum_{i=1}^{d} \mathbb{E}[\hat{v}_{T,i}^{1/2}]}{\alpha(1-\beta_1)\sqrt{T}} + \frac{D_\infty^2}{2(1-\beta_1)T} \sum_{t=1}^{T}\sum_{i=1}^{d} \frac{\beta_1 \mathbb{E}[\hat{v}_{t,i}^{1/2}]}{\alpha\sqrt{t}} + \frac{\alpha\sqrt{1+\log T}\sum_{i=1}^{d} \mathbb{E}\|\hat{\mathbf{g}}_{1:T,i}\|}{(1-\beta_1)^2(1-\gamma)\sqrt{1-\beta_2}T}. \tag{16}$$

*where $f_{t,\mu}$ denotes the smoothing function of $f$ defined by* (2), *$\hat{v}_{t,i}$ denotes the $i$th element of the vector $\hat{\mathbf{v}}_t$ defined in Algorithm 1, and $\hat{\mathbf{g}}_{1:T,i} := [\hat{g}_{1,i}, \ldots, \hat{g}_{T,i}]^\top$.*

We remark that Proposition 4 would reduce to [18, Theorem 4] by replacing ZO gradient estimates $\hat{\mathbf{g}}_{1:T,i}$ and $\hat{v}_{t,i}$ with FO gradients $\mathbf{g}_{1:T}$ and $v_t$. However, it was recently shown by [39] that the proof of [18, Theorem 4] is problematic. To address the proof issue, in Proposition 4 we present a simpler fix than [39, Theorem 4.1] and show that the conclusion of [18, Theorem 4] keeps correct. In the FO setting, the rate of AdaMM under **A2** for constrained convex optimization is given by $O(d/\sqrt{T})$ [19, Corollary 4.4]. Here **A2** provides the direct $\eta$-upper bound on $|g_{t,i}|$ and $\hat{v}_{t,i}^{1/2}$, and we consider worst-case rate analysis without imposing extra assumptions like sparse gradients[3]. In the ZO setting, we need further bound $|\hat{g}_{t,i}|$ and $\hat{v}_{t,i}$ and link $R_{T,\mu}$ to $R_T$, where the former is achieved by Proposition 3 and the latter is achieved by the relationship between $f_t$ and its smoothing function $f_{t,\mu}$ shown in Lemma A1-(a), yielding $f_t(\mathbf{x}_t) - f_t(\mathbf{x}^*) \leq f_{t,\mu}(\mathbf{x}_t) - f_{t,\mu}(\mathbf{x}^*) + 2\mu L_c$. Thus, given $\mu \leq d/\sqrt{T}$ and assuming conditions in Proposition 3 hold, then the rate of ZO-AdaMM becomes $R_T \leq 2\mu L_c + R_{T,\mu} = O(d^{1.5}/\sqrt{T})$, which is $O(\sqrt{d})$ worse than the AdaMM.

**Comparison with other ZO methods**   Since the existing convergence analysis for different ZO methods is built on different problem settings and assumptions. The direct comparison over the convergence rates might not be fair enough. Thus, in Table 1 we compare ZO-AdaMM with others ZO methods from 4 perspectives: a) the type of gradient estimator, b) the setting of smoothing parameter $\mu$, c) convergence rate, and d) function query complexity.

Table 1 shows that for unconstrained nonconvex optimization, the convergence of ZO-AdaMM achieves worse dependency on $d$ than ZO-SGD [9], ZO-SCD [22] and ZO-signSGD [14]. However, it has milder choice of $\mu$ than ZO-SGD, less query complexity than ZO-SCD, and no $T$-independent convergence bias compared to ZO-signSGD. Also, for constrained nonconvex optimization, ZO-AdaMM yields the similar rate to ZO-ProxSGD [27], which also implies ZO projected SGD (ZO-PSGD). For constrained convex optimization, the rate of ZO-AdaMM is $O(d)$ worse than ZO-SMD [23] but ours has the significantly improved dimension-dependency in $\mu$. We also highlight that at the first glance, ZO-AdaMM has a worse $d$-dependency (regardless of choice of $\mu$) than ZO-SGD. However, even in the FO setting, AdaMM has an extra $O(\sqrt{d})$ dependency in the worst case due to the effect of (coordinate-wise) gradient normalization when bounding the distance of two consecutive

updates. Thus, in addition to comparing with different ZO methods, Table 1 also summarizes the convergence performance of FO AdaMM. Note that our rate yields $O(\sqrt{d})$ slowdown compared to FO AdaMM though bounding ZO gradient estimate norm requires stricter assumption.

| Method | Assumptions | Gradient estimator | Smoothing parameter $\mu$ | Rate | Query |
|---|---|---|---|---|---|
| ZO-SGD [9] | NC[1], UCons[1], **A1**, **A3**[2] | GauGE[1] | $O\left(\frac{1}{d\sqrt{T}}\right)$ | $O\left(\frac{\sqrt{d}}{\sqrt{T}} + \frac{d}{T}\right)$ | $O(T)$ |
| ZO-SCD [22] | NC, UCons, **A1**, **A3**[2] | CooGE[1] | $O\left(\frac{1}{\sqrt{T}} + \frac{1}{\sqrt{d}}\right)$ | $O\left(\frac{\sqrt{d}}{\sqrt{T}} + \frac{d}{T}\right)$ | $O(dT)$ |
| ZO-signSGD [14] | NC, UCons, **A1**, **A3** | sign-UniGE[1] | $O\left(\frac{1}{\sqrt{dT}}\right)$ | $O(\frac{\sqrt{d}}{\sqrt{T}} + \frac{\sqrt{d}}{\sqrt{b}} + \frac{d}{\sqrt{bq}})$[3] | $O(bqT)$ |
| ZO-ProxSGD / ZO-PSGD [27] | NC, Cons[4], **A1**, **A3** | GauGE | $O\left(\frac{1}{\sqrt{dT}}\right)$ | $O\left(\frac{d^2}{qT} + \frac{d}{q}\right)$ | $O(qT)$ |
| ZO-SMD [23] | C, Cons, **A3** | GauGE/UniGE | $O\left(\frac{1}{dt}\right)$ | $O\left(\frac{\sqrt{d}}{\sqrt{T}}\right)$ | $O(T)$ |
| *AdaMM* [20, 38] | NC, UCons, **A1**, **A2** | SGE[1] | n/a | $O\left(\frac{\sqrt{d}}{\sqrt{T}} + \frac{d}{T}\right)$ | n/a |
| *AdaMM* [18, 19, 39] | C, Cons, **A2** | SGE | n/a | $O\left(\frac{d}{\sqrt{T}}\right)$ | n/a |
| **ZO-AdaMM** | NC, UCons, **A1**, **A3** | UniGE | $O\left(\frac{1}{\sqrt{dT}}\right)$ | $O\left(\frac{d}{\sqrt{T}} + \frac{d^2}{T}\right)$ | $O(T)$ |
| **ZO-AdaMM** | NC, Cons, **A1**, **A3** $\beta_{1,t}=0$ | UniGE | $O\left(\frac{1}{\sqrt{dT}}\right)$ | $O\left(\frac{d}{T} + \frac{1}{b} + \frac{d}{q}\right)$ | $O(bqT)$ |
| **ZO-AdaMM** | C, Cons, **A3** | UniGE | $O\left(\frac{d}{\sqrt{T}}\right)$ | $O\left(\frac{d^{1.5}}{\sqrt{T}}\right)$ | $O(T)$ |

[1] *Abbreviations*. NC: Nonconvex; UCons: Unconstrained; GauGE: Gaussian random vector based gradient estimate; UniGE: Uniform random vector based gradient estimate; CooGE: Coordinate-wise gradient estimate; SGE: stochastic (first-order) gradient estimate

[2] Assumption of bounded variance of stochastic gradients is implied from **A3**.

[3] Convergence of ZO-signSGD is measured by $\mathbb{E}[\|\nabla f(\mathbf{x}_T)\|_2]$ rather than its square used in other algorithms for nonconvex optimization.

**Table 1:** Summary of convergence rate and query complexity of various ZO algorithms given $T$ iterations.

## 6 Applications to Black-Box Adversarial Attacks

In this section, we demonstrate the effectiveness of ZO-AdaMM by experiments on generating black-box adversarial examples. Our experiments will be performed on Inception V3 [45] using ImageNet [46]. Here we focus on two types of black-box adversarial attacks: *per-image* adversarial perturbation [47] and *universal* adversarial perturbation against multiple images [5, 6, 48, 49]. For each type of attack, we allow both constrained and unconstrained optimization problem settings. We compare our propos ed ZO-AdaMM method with 6 existing ZO algorithms: ZO-SGD, ZO-SCD and ZO-signSGD for unconstrained optimization, and ZO-PSGD, ZO-SMD and ZO-NES for constrained optimization. The first 5 methods have been summarized in Table 1, and ZO-NES refers to the black-box attack generation method in [6], which applies a projected version of ZO-signSGD using natural evolution strategy (NES) based random gradient estimator. In our experiments, every method takes the same number of queries per iteration. Accordingly, the total query complexity is consistent with the number of iterations. We refer to Appendix 4 for details on experiment setups.

**Per-image adversarial perturbation** In Fig. 1, we present the attack loss and the resulting $\ell_2$-distortion against iteration numbers for solving both unconstrained and constrained adversarial attack problems, namely, (94) and (93) in Appendix 4, over 100 randomly selected images. Here every algorithm is initialized by zero perturbation. Thus, as the iteration increases, the attack loss decreases until it converges to 0 (indicating successful attack) while the distortion could increase. At this sense, the best attack performance should correspond to the best tradeoff between the fast convergence to 0 attack loss and the low distortion power (evaluated by $\ell_2$ norm). As we can see, ZO-AdaMM consistently outperforms other ZO methods in terms of the fast convergence of attack loss and relatively small perturbation. We also note that ZO-signSGD and ZO-NES have poor convergence accuracy in terms of either large attack loss or large distortion at final iterations. This is not surprising, since it has been shown in [14] that ZO-signSGD only converges to a neighborhood of a solution, and ZO-NES can be regarded as a Euclidean projection based ZO-signSGD, which could induce convergence issues shown by Prop. 1. We refer readers to Table A3 for detailed experiment results.

**Universal adversarial perturbation** We now focus on designing a universal adversarial perturbation using the constrained attack problem formulation. Here we attack $M = 100$ random selected images from ImageNet. In Fig. 2, we present the attack loss as well as the $\ell_2$ norm of universal perturbation at different iteration numbers. As we can see, compared with the other ZO algorithms, ZO-AdaMM has the fastest convergence speed to reach the smallest adversarial perturbation (namely, strongest universal attack). Moreover, in Table 2 we present detailed attack success rate and $\ell_2$ distortion over $T = 40000$ iterations. Consistent with Fig. 2, ZO-AdaMM achieves highest success rate

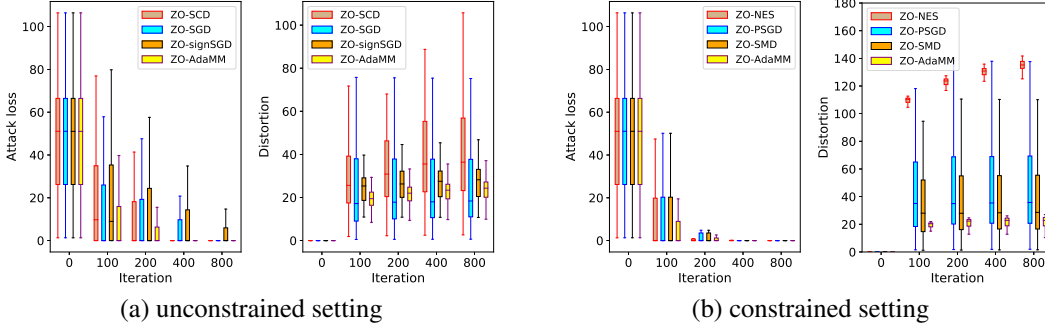

(a) unconstrained setting                    (b) constrained setting

**Figure 1:** The attack loss and adversarial distortion v.s. iterations. Each box represents results from 100 images.

with lowest distortion. In Fig. A2 of Appendix A2, we visualize patterns of the generated universal adversarial perturbations which further confirm the advantage of ZO-AdaMM.

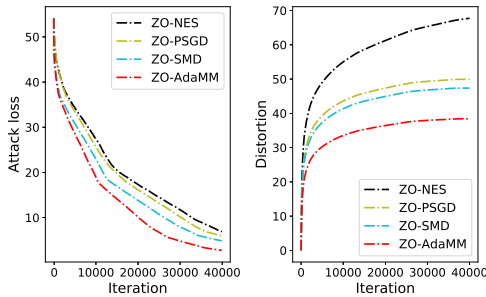

**Figure 2:** Attack loss and distortion of universal attack.

| Methods | Attack success rate | Final $\|\boldsymbol{\delta}_T\|_2^2$ |
|---------|---------------------|---------------------------------------|
| ZO-NES | 74% | 67.74 |
| ZO-PSGD | 78% | 49.92 |
| ZO-SMD | 79% | 47.36 |
| ZO-AdaMM | **84%** | **38.40** |

**Table 2:** Summary of attack success rate and eventual $\ell_2$ distortion for universal attack against 100 images under $T = 40000$ iterations.

## 7   Conclusion

In this paper, we propose ZO-AdaMM, the first effort to integrate adaptive momentum methods with ZO optimization. In theory, we show that ZO-AdaMM has convergence guarantees for both convex and nonconvex constrained optimization. Compared with (first-order) AdaMM, it suffers a slowdown factor of $O(\sqrt{d})$. Particularly, we establish a new Mahalanobis distance based convergence measure whose necessity and importance are provided in characterizing the convergence behavior of ZO-AdaMM on nonconvex constrained problems. To demonstrate the utility of the algorithm, we show the superior performance of ZO-AdaMM for designing adversarial examples from black-box neural networks. Compared with 6 state-of-the-art ZO methods, ZO-AdaMM has the fastest empirical convergence to strong black-box adversarial attacks that require the minimum distortion strength.

## Acknowledgement

This work is partly supported by National Science Foundation CNS-1932351. M. Hong is supported in part by NSF under Grant CMMI-172775, CIF-1910385 and by ARO under grant 73202-CS.

## Footnotes

[2]In the paper, we could omit $\log(T)$ in Big $O$ notation.

[3]The work [40] showed the lack of sparsity in gradients while generating adversarial examples.

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
