[Supplementary Material]

# Appendix

## 1 Smoothing Function and Random Gradient Estimate

**Lemma A1** *a) Relationship between $f_\mu$ and $f$: If $f$ is convex, then $f_\mu$ is convex. If $f$ is $L_c$-Lipschitz continuous, then $f_\mu$ is $L_c$-Lipschitz continuous. Moreover for any $\mathbf{x} \in \mathbb{R}^d$,*

$$|f_\mu(\mathbf{x}) - f(\mathbf{x})| \leq L_c \mu. \tag{17}$$

*If $f$ has $L_g$-Lipschitz continuous gradient, then $f_\mu$ has $L_g$-Lipschitz continuous gradient. Moreover for any $\mathbf{x} \in \mathbb{R}^d$,*

$$|f_\mu(\mathbf{x}) - f(\mathbf{x})| \leq L_g \mu^2 / 2 \tag{18}$$

$$\|\nabla f_\mu(\mathbf{x}) - \nabla f(\mathbf{x})\|_2^2 \leq \mu^2 d^2 L_g^2 / 4. \tag{19}$$

*b) Statistical properties of $\hat{\nabla} f$: For any $\mathbf{x} \in \mathbb{R}^d$,*

$$\mathbb{E}_\mathbf{u} \left[ \hat{\nabla} f(\mathbf{x}) \right] = \nabla f_\mu(\mathbf{x}). \tag{20}$$

*If $f$ has $L_g$-Lipschitz continuous gradient, then*

$$\mathbb{E}_\mathbf{u} \left[ \|\hat{\nabla} f(\mathbf{x})\|_2^2 \right] \leq 2d \|\nabla f(\mathbf{x})\|_2^2 + \mu^2 L_g^2 d^2 / 2. \tag{21}$$

**Proof:** We refer readers to [30, Lemma 4.1] for the detailed proof of a)-b) except the Lipschitz continuity of $f_\mu$ and (17). Suppose that $f$ is $L_c$-Lipschitz continuous, based on the definition of $f_\mu$ in (2), we obtain

$$|f_\mu(\mathbf{x}) - f_\mu(\mathbf{y})| \leq \frac{1}{\alpha(d)} \int_B |f(\mathbf{x} + \mu\mathbf{u}) - f(\mathbf{y} + \mu\mathbf{u})| d\mathbf{u} \leq L_c \|\mathbf{x} - \mathbf{y}\|_2,$$

where $\alpha(d)$ denotes the volume of the unit ball $B$ in $\mathbb{R}^d$.

Moreover, we prove (17) as below.

$$|f_\mu(\mathbf{x}) - f(\mathbf{x})| = \left| \frac{1}{\alpha(d)} \int_B f(\mathbf{x} + \mu\mathbf{u}) - f(\mathbf{x}) d\mathbf{u} \right| \leq \frac{\mu L_c}{\alpha(d)} \int_B \|\mathbf{u}\|_2 d\mathbf{u} = \frac{\mu L_c d}{d+1} \leq \mu L_c,$$

where the first equality holds due to (2), Jensen's inequality and Lipschitz continuity of $f$, and the last equality holds since $(1/\alpha(d)) \int_B \|\mathbf{u}\|_2^p d\mathbf{u} = \frac{n}{n+p}$ [30, Lemma 6.3.a]. $\qquad\square$

In Lemma A1, it is clear from (19) and (20) that the ZO gradient estimate (1) becomes unbiased to the true gradient $\nabla f$ only when $\mu \to 0$. However, if $\mu$ is too small, then the difference of empirical function values is also too small to represent the function differential [22, 24]. Thus, the tolerance on the smoothing parameter $\mu$ is an important factor to indicate the convergence performance of ZO optimization methods. It is also known from (21) that regardless of the value of $\mu$, the variance of the ZO gradient estimate is always proportional to the dimension $d$. This is one of reasons for the dimension-dependent slowdown in convergence of ZO optimization methods. This also introduces technical difficulties for analyzing the effect of adaptive learning rate on the convergence of ZO-AdaMM in nonconvex optimization.

## 2 Proof for Nonconvex Optimization

### 2.1 Proof of Proposition 1

Let us consider a special case of Algorithm 1 with the average ZO gradient estimate $\hat{\nabla} f(\mathbf{x}) = \frac{d}{q\mu} \sum_{i=1}^q \{[f(\mathbf{x} + \mu\mathbf{u}_i) - f(\mathbf{x})]\mathbf{u}_i\}$ under $\beta_{1,t} = \beta_2 \to 0$, $\mu \to 0$ and $q \to \infty$. The conditions of $\beta_{1,t} = \beta_2 \to 0$ enables Algorithm 1 to reduce to ZO-signSGD in [14], and the conditions of $\mu \to 0$ and $q \to \infty$ makes the ZO gradient estimate unbiased to $\nabla f(\mathbf{x})$ and its variance close to 0 [14, Proposition 2]. As a result, we obtain $\hat{\mathbf{g}}_t \to \nabla f(\mathbf{x}_t)$, and Algorithm 1 becomes signSGD [15],

$$\mathbf{x}_{t+1} = \Pi_{\mathcal{X}, \mathbf{I}}(\mathbf{x}_t - \alpha_t \text{sign}(\nabla f(\mathbf{x}_t))) \tag{22}$$

where $\text{sign}(x) = 1$ if $x > 0$ and $-1$ if $x < 0$, and it is taken elementwise for a vector argument.

Let $f(\mathbf{x}) = -2x_1 - x_2$ in (5). We then run (22) at $x_1 = x_2 = 0.5$, which yields

$$\mathbf{x}_{t+1} = \Pi_{\mathcal{X}}([0.5, 0.5]^T - \alpha_t[-1, -1]^T) = \Pi_X([0.5 + \alpha_t, 0.5 + \alpha_t]^T) = [0.5, 0.5]^T, \tag{23}$$

where $\mathcal{X}$ encodes the constraint $|x_1 + x_2| \leq 1$.

It is clear that the updating rule (23) will converge to $\mathbf{x} = [0.5, 0.5]^T$ regardless of the choice of $\alpha_t$. The remaining question is whether or not it is a stationary point. Recall that a point $\mathbf{x}^*$ is a stationary point if it satisfies the following conditions:

$$\langle \nabla f(\mathbf{x}^*), \mathbf{x} - \mathbf{x}^* \rangle \geq 0, \ \forall \mathbf{x} \in \mathcal{X}. \tag{24}$$

Since the gradient at $[0.5, 0.5]^T$ is $[-2, -1]^T$, and the inequality (24) at $\mathbf{x} = [0.6, 0.4]^T \in X$ does *not* hold, given by $\langle [-2, -1]^T, [0.6, 0.4]^T - [0.5, 0.5]^T \rangle = -0.1 < 0$. This implies that $\mathbf{x}^* = [0.5, 0.5]^T$ is *not* a stationary point of problem (5).

Next, we apply the Mhalanobis distance $\hat{\mathbf{V}}_t = \mathrm{diag}(\nabla f(\mathbf{x}_t)^2)$ to (22),

$$\mathbf{x}_{t+1} = \Pi_{\mathcal{X}, \hat{\mathbf{V}}_t^{1/2}}(\mathbf{x}_t - \alpha_t \mathrm{sign}(\nabla f(\mathbf{x}_t))) = \Pi_{\mathcal{X}, \hat{\mathbf{V}}_t^{1/2}}(\mathbf{x}_t - \alpha_t \hat{\mathbf{V}}_t^{-1/2} \nabla f(\mathbf{x}_t)). \tag{25}$$

Similar to (22), we then consider the impact of fixed point $\mathbf{x}_{t+1} = \mathbf{x}_t$ on (25). By the definition of projection operator, we have

$$\mathbf{x}_t = \arg\min_{\mathbf{x} \in \mathcal{X}} \|\hat{\mathbf{V}}^{1/4}(\mathbf{x} - \mathbf{x}_t + \alpha_t \hat{\mathbf{V}}_t^{-1/2} \nabla f(\mathbf{x}_t)))\| \tag{26}$$

The optimality condition of (26) is given by

$$\langle \hat{\mathbf{V}}^{1/2}(\mathbf{x}_t - \mathbf{x}_t + \alpha_t \hat{\mathbf{V}}_t^{-1/2} \nabla f(\mathbf{x}_t)), \mathbf{x} - \mathbf{x}_t \rangle \geq 0, \ \forall x \in X,$$

which reduces to

$$\langle \nabla f(\mathbf{x}_t), \mathbf{x} - \mathbf{x}_t \rangle \geq 0, \ \forall \mathbf{x} \in X. \tag{27}$$

It thus means that $\mathbf{x}_t$ is a stationary point by (24). $\square$

## 2.2 Proof of Proposition 2

Before proving the main result Proposition 2, we first prove a few auxiliary lemmas.

**Lemma 2.1** *Given $\{\mathbf{x}_t\}$ from Algorithm 1, consider the sequence*

$$\mathbf{z}_t = \mathbf{x}_t + \frac{\beta_1}{1 - \beta_1}(\mathbf{x}_t - \mathbf{x}_{t-1}), \ \forall t \geq 1, \tag{28}$$

*where let $\mathbf{x}_0 := \mathbf{x}_1$. Then for $\beta_{1,t} = \beta_1$ and $\mathcal{X} = \mathbb{R}^d$, $\forall t > 1$*

$$\mathbf{z}_{t+1} - \mathbf{z}_t$$
$$= -\frac{\beta_1}{1 - \beta_1}\left(\alpha_t \hat{\mathbf{V}}_t^{-1/2} - \alpha_{t-1} \hat{\mathbf{V}}_{t-1}^{-1/2}\right)\mathbf{m}_{t-1} - \alpha_t \hat{\mathbf{V}}_t^{-1/2} \hat{\mathbf{g}}_t$$

*and*

$$\mathbf{z}_2 - \mathbf{z}_1 = -\alpha_1 \hat{\mathbf{g}}_1 / \sqrt{\hat{\mathbf{v}}_1}.$$

**Proof of Lemma 2.1:** The proof follows from Lemma 6.1 in [20] by setting $\beta_{1,t} = \beta_1$.

**Lemma 2.2** *By ZO-AdaMM update rule, we have*

$$\mathbb{E}[f_\mu(\mathbf{z}_{t+1}) - f_\mu(\mathbf{z}_1)] \leq \sum_{t=1}^{T} E\left[\langle \nabla f_\mu(\mathbf{x}_t), \mathbf{z}_{t+1} - \mathbf{z}_t \rangle\right] + \frac{4L_g + 5L_g\beta_1^2}{2(1 - \beta_1)^2}\sum_{t=1}^{T} E\left[\|\mathbf{x}_{t+1} - \mathbf{x}_t\|^2\right] \tag{29}$$

**Proof of Lemma 2.2:** By smoothness of function $f$, we can have

$$f_\mu(\mathbf{z}_{t+1}) - f_\mu(\mathbf{z}_t)$$

$$\leq \langle \nabla f_\mu(\mathbf{z}_t), \mathbf{z}_{t+1} - \mathbf{z}_t \rangle + \frac{L_g}{2}\|\mathbf{z}_{t+1} - \mathbf{z}_t\|^2$$

$$= \langle \nabla f_\mu(\mathbf{x}_t), \mathbf{z}_{t+1} - \mathbf{z}_t \rangle + \frac{L_g}{2}\|\mathbf{z}_{t+1} - \mathbf{z}_t\|^2 + \langle \nabla f_\mu(\mathbf{z}_t) - \nabla f_\mu(\mathbf{x}_t), \mathbf{z}_{t+1} - \mathbf{z}_t \rangle$$

$$\leq \langle \nabla f_\mu(\mathbf{x}_t), \mathbf{z}_{t+1} - \mathbf{z}_t \rangle + \frac{L_g}{2}\|\mathbf{z}_{t+1} - \mathbf{z}_t\|^2 + \frac{1}{2}\left(\frac{1}{L_g}\|\nabla f_\mu(\mathbf{z}_t) - \nabla f_\mu(\mathbf{x}_t)\|^2 + L_g\|\mathbf{z}_{t+1} - \mathbf{z}_t\|^2\right)$$

$$\leq \langle \nabla f_\mu(\mathbf{x}_t), \mathbf{z}_{t+1} - \mathbf{z}_t \rangle + L_g\|\mathbf{z}_{t+1} - \mathbf{z}_t\|^2 + \frac{1}{2}L_g\|\mathbf{z}_t - \mathbf{x}_t\|^2$$

$$= \langle \nabla f_\mu(\mathbf{x}_t), \mathbf{z}_{t+1} - \mathbf{z}_t \rangle + L_g\|\mathbf{z}_{t+1} - \mathbf{z}_t\|^2 + \frac{1}{2}L_g\left\|\frac{\beta_1}{1-\beta_1}(\mathbf{x}_t - \mathbf{x}_{t-1})\right\|^2 \tag{30}$$

Further, by (28), we have

$$\mathbf{z}_{t+1} - \mathbf{z}_t = \frac{1}{1-\beta_1}(\mathbf{x}_{t+1} - \mathbf{x}_t) + \frac{\beta_1}{1-\beta_1}(\mathbf{x}_t - \mathbf{x}_{t-1})$$

and thus

$$\|\mathbf{z}_{t+1} - \mathbf{z}_t\|^2 \leq \frac{2}{(1-\beta_1)^2}\|\mathbf{x}_{t+1} - \mathbf{x}_t\|^2 + \frac{2\beta_1^2}{(1-\beta_1)^2}\|\mathbf{x}_t - \mathbf{x}_{t-1}\|^2 \tag{31}$$

Substituting (31) into (30), we get

$$f_\mu(\mathbf{z}_{t+1}) - f_\mu(\mathbf{z}_t)$$

$$\leq \langle \nabla f_\mu(\mathbf{x}_t), \mathbf{z}_{t+1} - \mathbf{z}_t \rangle + \frac{2L_g}{(1-\beta_1)^2}\|\mathbf{x}_{t+1} - \mathbf{x}_t\|^2 + \frac{2\beta_1^2 L_g}{(1-\beta_1)^2}\|\mathbf{x}_t - \mathbf{x}_{t-1}\|^2$$

$$+ \frac{1}{2}L_g\frac{\beta_1^2}{(1-\beta_1)^2}\|\mathbf{x}_t - \mathbf{x}_{t-1}\|^2$$

$$= \langle \nabla f_\mu(\mathbf{x}_t), \mathbf{z}_{t+1} - \mathbf{z}_t \rangle + \frac{2L_g}{(1-\beta_1)^2}\|\mathbf{x}_{t+1} - \mathbf{x}_t\|^2 + \frac{5L_g\beta_1^2}{2(1-\beta_1)^2}\|\mathbf{x}_t - \mathbf{x}_{t-1}\|^2 \tag{32}$$

Summing $t$ from 1 to $T$ and take expectation, we get

$$\mathbb{E}[f_\mu(\mathbf{z}_{t+1}) - f_\mu(\mathbf{z}_1)]$$

$$\leq \mathbb{E}\left[\sum_{t=1}^{T}\left(\langle \nabla f_\mu(\mathbf{x}_t), \mathbf{z}_{t+1} - \mathbf{z}_t \rangle + \frac{2L_g}{(1-\beta_1)^2}\|\mathbf{x}_{t+1} - \mathbf{x}_t\|^2 + \frac{5L_g\beta_1^2}{2(1-\beta_1)^2}\|\mathbf{x}_t - \mathbf{x}_{t-1}\|^2\right)\right]$$

$$\leq \sum_{t=1}^{T}\mathbb{E}\left[\langle \nabla f_\mu(\mathbf{x}_t), \mathbf{z}_{t+1} - \mathbf{z}_t \rangle\right] + \frac{4L_g + 5L_g\beta_1^2}{2(1-\beta_1)^2}\sum_{t=1}^{T}\mathbb{E}\left[\|\mathbf{x}_{t+1} - \mathbf{x}_t\|^2\right]$$

$\square$

**Lemma 2.3** *Assume* $\|\hat{g}_t\|_\infty \leq G_{zo}$, $\forall t \in [T]$ *and* $m_0 = 0$, *By ZO-AdaMM update rule, we have*

$$\sum_{t=1}^{T}\mathbb{E}[\langle \nabla f_\mu(\mathbf{x}_t), \mathbf{z}_{t+1} - \mathbf{z}_t \rangle] \leq \mathbb{E}\left[\left(\frac{\eta G_{zo}}{1-\beta_1} + \eta^2\right)\sum_{i=1}^{d}\frac{\alpha_1}{\sqrt{\hat{\mathbf{v}}_{0,i}}}\right]$$

$$- \sum_{t=1}^{T}\mathbb{E}\left[\langle \nabla f_\mu(\mathbf{x}_t), \alpha_t\hat{\mathbf{V}}_t^{-1/2}\nabla f_\mu(\mathbf{x}_t)\rangle\right]. \tag{33}$$

**Proof of Lemma 2.3:** By Lemma 2.1, we have

$$\langle \nabla f_\mu(\mathbf{x}_t), \mathbf{z}_{t+1} - \mathbf{z}_t \rangle$$

$$= \langle \nabla f_\mu(\mathbf{x}_t), -\frac{\beta_1}{1-\beta_1}\left(\alpha_t\hat{\mathbf{V}}_t^{-1/2} - \alpha_{t-1}\hat{\mathbf{V}}_{t-1}^{-1/2}\right)\mathbf{m}_{t-1} - \alpha_t\hat{\mathbf{V}}_t^{-1/2}\hat{\mathbf{g}}_t\rangle$$

$$= \langle \nabla f_\mu(\mathbf{x}_t), -\frac{\beta_1}{1-\beta_1}\left(\alpha_t\hat{\mathbf{V}}_t^{-1/2} - \alpha_{t-1}\hat{\mathbf{V}}_{t-1}^{-1/2}\right)\mathbf{m}_{t-1}\rangle - \langle \nabla f_\mu(\mathbf{x}_t), \alpha_t\hat{\mathbf{V}}_t^{-1/2}\hat{\mathbf{g}}_t\rangle, \tag{34}$$

and

$$\langle \nabla f_\mu(\mathbf{x}_t), \alpha_t \hat{\mathbf{V}}_t^{-1/2} \hat{\mathbf{g}}_t \rangle$$
$$=\langle \nabla f_\mu(\mathbf{x}_t), \alpha_t \hat{\mathbf{V}}_t^{-1/2} \nabla f_\mu(\mathbf{x}_t) \rangle + \langle \nabla f_\mu(\mathbf{x}_t), \alpha_t \hat{\mathbf{V}}_t^{-1/2} (\hat{\mathbf{g}}_t - \nabla f_\mu(\mathbf{x}_t)) \rangle$$
$$=\langle \nabla f_\mu(\mathbf{x}_t), \alpha_t \hat{\mathbf{V}}_t^{-1/2} \nabla f_\mu(\mathbf{x}_t) \rangle + \langle \nabla f_\mu(\mathbf{x}_t), \alpha_{t-1} \hat{\mathbf{V}}_{t-1}^{-1/2} (\hat{\mathbf{g}}_t - \nabla f_\mu(\mathbf{x}_t)) \rangle$$
$$+ \langle \nabla f_\mu(\mathbf{x}_t), \left( \alpha_t \hat{\mathbf{V}}_t^{-1/2} - \alpha_{t-1} \hat{\mathbf{V}}_{t-1}^{-1/2} \right) (\hat{\mathbf{g}}_t - \nabla f_\mu(\mathbf{x}_t)) \rangle. \tag{35}$$

Substitute (35) into (34), we have

$$\langle \nabla f_\mu(\mathbf{x}_t), \mathbf{z}_{t+1} - \mathbf{z}_t \rangle$$
$$\leq \langle \nabla f_\mu(\mathbf{x}_t), -\frac{\beta_1}{1-\beta_1} \left( \alpha_t \hat{\mathbf{V}}_t^{-1/2} - \alpha_{t-1} \hat{\mathbf{V}}_{t-1}^{-1/2} \right) \odot \mathbf{m}_{t-1} \rangle$$
$$- \langle \nabla f_\mu(\mathbf{x}_t), \alpha_t \hat{\mathbf{V}}_t^{-1/2} \nabla f_\mu(\mathbf{x}_t) \rangle - \langle \nabla f_\mu(\mathbf{x}_t), \alpha_{t-1} \hat{\mathbf{V}}_{t-1}^{-1/2} (\hat{\mathbf{g}}_t - \nabla f_\mu(\mathbf{x}_t)) \rangle$$
$$- \langle \nabla f_\mu(\mathbf{x}_t), \left( \alpha_t \hat{\mathbf{V}}_t^{-1/2} - \alpha_{t-1} \hat{\mathbf{V}}_{t-1}^{-1/2} \right) (\hat{\mathbf{g}}_t - \nabla f_\mu(\mathbf{x}_t)) \rangle$$
$$= \langle \nabla f_\mu(\mathbf{x}_t), - \left( \alpha_t \hat{\mathbf{V}}_t^{-1/2} - \alpha_{t-1} \hat{\mathbf{V}}_{t-1}^{-1/2} \right) \frac{\mathbf{m}_t}{1-\beta_1} \rangle$$
$$- \langle \nabla f_\mu(\mathbf{x}_t), - \left( \alpha_t \hat{\mathbf{V}}_t^{-1/2} - \alpha_{t-1} \hat{\mathbf{V}}_{t-1}^{-1/2} \right) \nabla f_\mu(\mathbf{x}_t) \rangle$$
$$- \langle \nabla f_\mu(\mathbf{x}_t), \alpha_t \hat{\mathbf{V}}_t^{-1/2} \nabla f_\mu(\mathbf{x}_t) \rangle - \langle \nabla f_\mu(\mathbf{x}_t), \alpha_{t-1} \hat{\mathbf{V}}_{t-1}^{-1/2} (\hat{\mathbf{g}}_t - \nabla f_\mu(\mathbf{x}_t)) \rangle$$
$$\leq (\frac{\eta G_{zo}}{1-\beta_1} + \eta^2) \sum_{i=1}^{d} \left| \frac{\alpha_{t-1}}{\sqrt{\hat{\mathbf{v}}_{t-1,i}}} - \frac{\alpha_t}{\sqrt{\hat{\mathbf{v}}_{t,i}}} \right|$$
$$- \langle \nabla f_\mu(\mathbf{x}_t), \alpha_t \hat{\mathbf{V}}_t^{-1/2} \nabla f_\mu(\mathbf{x}_t) \rangle - \langle \nabla f_\mu(\mathbf{x}_t), \alpha_{t-1} \hat{\mathbf{V}}_{t-1}^{-1/2} (\hat{\mathbf{g}}_t - \nabla f_\mu(\mathbf{x}_t)) \rangle \tag{36}$$

where the last inequality follows from the assumption that $\hat{\mathbf{V}}_t = \text{diga}(\hat{\mathbf{v}}_t)$, $\|\nabla f_\mu(\mathbf{x}_t)\|_\infty \leq \eta$ and $\|\hat{\mathbf{g}}_t\|_\infty \leq G_{zo}$.

The upper bound on $\|\mathbf{m}_t\|_\infty$ can be proved by a simple induction. Recall that $\mathbf{m}_t = \beta_{1,t} \mathbf{m}_{t-1} + (1-\beta_{1,t}) \hat{\mathbf{g}}_t$, suppose $\|\mathbf{m}_{t-1}\| \leq G_{zo}$, we have

$$\|\mathbf{m}_t\|_\infty \leq (\beta_{1,t} + (1-\beta_{1,t})) \max(\|\hat{\mathbf{g}}_t\|_\infty, \|\mathbf{m}_{t-1}\|_\infty)$$
$$= \max(\|\hat{\mathbf{g}}_t\|_\infty, \|\mathbf{m}_{t-1}\|_\infty) \leq G_{zo}. \tag{37}$$

Then since $\mathbf{m}_0 = 0$, we have $\|\mathbf{m}_0\| \leq G_{zo}$, which completes the induction.

Sum $t$ from 1 to $T$ and take expectation over randomness of $\hat{\mathbf{g}}_t$, we have

$$\sum_{t=1}^{T} \mathbb{E}[\langle \nabla f_\mu(\mathbf{x}_t), \mathbf{z}_{t+1} - \mathbf{z}_t \rangle]$$
$$\leq \mathbb{E}\left[ \sum_{t=1}^{T} \left( \frac{\eta G_{zo}}{1-\beta_1} + \eta^2 \right) \sum_{i=1}^{d} \left| \frac{\alpha_{t-1}}{\sqrt{\hat{\mathbf{v}}_{t-1,i}}} - \frac{\alpha_t}{\sqrt{\hat{\mathbf{v}}_{t,i}}} \right| \right]$$
$$- \sum_{t=1}^{T} \mathbb{E}\left[ \langle \nabla f_\mu(\mathbf{x}_t), \alpha_t \hat{\mathbf{V}}_t^{-1/2} \nabla f_\mu(\mathbf{x}_t) \rangle \right] - \sum_{t=1}^{T} \mathbb{E}\left[ \langle \nabla f_\mu(\mathbf{x}_t), \alpha_{t-1} \hat{\mathbf{V}}_{t-1}^{-1/2} (\hat{\mathbf{g}}_t - \nabla f_\mu(\mathbf{x}_t)) \rangle \right]$$
$$\leq \mathbb{E}\left[ \left( \frac{\eta G_{zo}}{1-\beta_1} + \eta^2 \right) \sum_{i=1}^{d} \frac{\alpha_1}{\sqrt{\hat{\mathbf{v}}_{0,i}}} \right] - \sum_{t=1}^{T} \mathbb{E}\left[ \langle \nabla f_\mu(\mathbf{x}_t), \alpha_t \hat{\mathbf{V}}_t^{-1/2} \nabla f_\mu(\mathbf{x}_t) \rangle \right]$$

where the last inequality follows from following facts.

1. Since $\hat{\mathbf{v}}_t = \max(\hat{\mathbf{v}}_{t-1}, \mathbf{v}_t)$, we know $\hat{\mathbf{v}}_t$ is non-decreasing. Given the fact that $\alpha_t$ is non-increasing (by our choice), we have $\alpha_{t-1}/\hat{\mathbf{v}}_{t-1,i} - \alpha_t/\hat{\mathbf{v}}_{t,i} \geq 0$. Thus, following inequality holds.

$$\mathbb{E}\left[\sum_{t=1}^{T}\sum_{i=1}^{d}\left|\frac{\alpha_{t-1}}{\sqrt{\hat{\mathbf{v}}_{t-1,i}}} - \frac{\alpha_t}{\sqrt{\hat{\mathbf{v}}_{t,i}}}\right|\right] = \mathbb{E}\left[\sum_{i=1}^{d}\sum_{t=1}^{T}\left|\frac{\alpha_{t-1}}{\sqrt{\hat{\mathbf{v}}_{t-1,i}}} - \frac{\alpha_t}{\sqrt{\hat{\mathbf{v}}_{t,i}}}\right|\right]$$

$$= \mathbb{E}\left[\sum_{i=1}^{d}\sum_{t=1}^{T}\left(\frac{\alpha_{t-1}}{\sqrt{\hat{\mathbf{v}}_{t-1,i}}} - \frac{\alpha_t}{\sqrt{\hat{\mathbf{v}}_{t,i}}}\right)\right] \leq \mathbb{E}\left[\sum_{i=1}^{d}\frac{\alpha_1}{\sqrt{\hat{\mathbf{v}}_{0,i}}}\right] \tag{38}$$

2. We have $\mathbb{E}[\hat{\mathbf{g}}_t|\hat{\mathbf{g}}_{1:t-1}] = \nabla f_\mu(\mathbf{x}_t)$ by the assumption that $\mathbb{E}[\hat{\mathbf{g}}_t] = \nabla f_\mu(\mathbf{x}_t)$ and the noise on $\hat{\mathbf{g}}_t$ is independent of $\hat{\mathbf{g}}_{1:t-1}$. Thus, the following holds

$$\mathbb{E}\left[\langle \nabla f_\mu(\mathbf{x}_t), \alpha_{t-1}\hat{\mathbf{V}}_{t-1}^{-1/2}(\hat{\mathbf{g}}_t - \nabla f_\mu(\mathbf{x}_t))\rangle\right] = 0 \tag{39}$$

$\square$

**Lemma 2.4** *Assume $\gamma := \beta_1/\beta_2 < 1$, ZO-AdaMM yields*

$$\|\mathbf{x}_{t+1} - \mathbf{x}_t\|^2 \leq \alpha_t^2 d\frac{1-\beta_1}{1-\beta_2}\frac{1}{1-\gamma} \tag{40}$$

**Comment:**This is an important lemma for ZO-AdaMM, it shows the squared update quantity is not dependent on size of stochastic gradient, thus giving a tighter dependency on $d$ compared with [18].

**Proof of Lemma 2.4:** By the update rule, we have

$$\|\mathbf{x}_{t+1} - \mathbf{x}_t\|^2 = \alpha_t^2\left\|\frac{\mathbf{m}_t}{\sqrt{\hat{\mathbf{v}}_t}}\right\|^2$$

$$\leq \alpha_t^2 \sum_{i=1}^{d}\frac{((1-\beta_1)\sum_{j=0}^{t-1}\beta_1^{t-j}\hat{\mathbf{g}}_{j,i})^2}{(1-\beta_2)\sum_{j=0}^{t-1}\beta_2^{t-j}\hat{\mathbf{g}}_{j,i}^2} \leq \alpha_t^2 \sum_{i=1}^{d}\frac{(1-\beta_1)^2(\sum_{j=0}^{t-1}\beta_1^{t-j})(\sum_{j=0}^{t-1}\beta_1^{t-j}\hat{\mathbf{g}}_{j,i}^2)}{(1-\beta_2)\sum_{j=0}^{t-1}\beta_2^{t-j}\hat{\mathbf{g}}_{j,i}^2}$$

$$\leq \alpha_t^2 \sum_{i=1}^{d}\frac{(1-\beta_1)\sum_{j=0}^{t-1}\beta_1^{t-j}\hat{\mathbf{g}}_{j,i}^2}{(1-\beta_2)\sum_{j=0}^{t-1}\beta_2^{t-j}\hat{\mathbf{g}}_{j,i}^2} \leq \alpha_t^2\sum_{i=1}^{d}\sum_{j=0}^{t-1}\frac{(1-\beta_1)\beta_1^{t-j}\hat{\mathbf{g}}_{j,i}^2}{(1-\beta_2)\beta_2^{t-j}\hat{\mathbf{g}}_{j,i}^2}$$

$$\leq \alpha_t^2 d\sum_{j=0}^{t-1}\frac{1-\beta_1}{1-\beta_2}\gamma^{t-j} \leq \alpha_t^2 d\frac{1-\beta_1}{1-\beta_2}\frac{1}{1-\gamma}$$

where the second inequality is due to Cauchy-Schwarz and $\gamma = \beta_1/\beta_2 < 1$.

$\square$

**Proof of Proposition 2:** Substitute (40) and (33) into (29), we get

$$\mathbb{E}[f_\mu(\mathbf{z}_{t+1}) - f_\mu(\mathbf{z}_1)]$$

$$\leq \sum_{t=1}^{T}\mathbb{E}[\langle \nabla f_\mu(\mathbf{x}_t), \mathbf{z}_{t+1} - \mathbf{z}_t\rangle] + \frac{4L_g + 5L_g\beta_1^2}{2(1-\beta_1)^2}\sum_{t=1}^{T}\mathbb{E}[\|\mathbf{x}_{t+1} - \mathbf{x}_t\|^2]$$

$$\leq \mathbb{E}\left[\left(\frac{\eta G_{zo}}{1-\beta_1} + \eta^2\right)\left\|\frac{\alpha_1}{\sqrt{\hat{\mathbf{v}}_0}}\right\|_1\right] - \sum_{t=1}^{T}\mathbb{E}\left[\langle \nabla f_\mu(\mathbf{x}_t), \alpha_t\hat{\mathbf{V}}_t^{-1/2}\nabla f_\mu(\mathbf{x}_t)\rangle\right]$$

$$+ \sum_{t=1}^{T}\alpha_t^2 d\frac{4L_g + 5L_g\beta_1^2}{2(1-\beta_1)^2}\frac{1-\beta_1}{1-\beta_2}\frac{1}{1-\gamma} \tag{41}$$

Rearrange and assume $f_\mu(\mathbf{z}_1) - \min_\mathbf{z} f_\mu(\mathbf{z}) \leq D_f$, we get

$$\sum_{t=1}^{T}\mathbb{E}\left[\langle \nabla f_\mu(\mathbf{x}_t), \alpha_t\hat{\mathbf{V}}_t^{-1/2}\nabla f_\mu(\mathbf{x}_t)\rangle\right]$$

$$\leq D_f + \mathbb{E}\left[\left(\frac{\eta G_{zo}}{1-\beta_1} + \eta^2\right)\left\|\frac{\alpha_1}{\sqrt{\hat{\mathbf{v}}_0}}\right\|_1\right] + \sum_{t=1}^{T}\alpha_t^2 d\frac{4L_g + 5L_g\beta_1^2}{2(1-\beta_1)^2}\frac{1-\beta_1}{1-\beta_2}\frac{1}{1-\gamma} \tag{42}$$

Set $\alpha_t = \alpha = 1/\sqrt{Td}$ and divide both sides by $T\alpha$, uniformly randomly pick $R$ from 1 to $T$,

$$\mathbb{E}\left[\|\hat{\mathbf{V}}_R^{-1/2}\nabla f_\mu(x_R)\|^2\right] = \frac{1}{T}\sum_{t=1}^{T}\mathbb{E}\left[\|V_t^{-1/2}\nabla f_\mu(\mathbf{x}_t)\|^2\right]$$

$$\leq\frac{D_f}{T\alpha} + \frac{1}{T}\mathbb{E}\left[\left(\frac{\eta G_{zo}}{1-\beta_1} + \eta^2\right)\left\|\frac{1}{\sqrt{\hat{\mathbf{v}}_0}}\right\|_1\right] + \alpha d\frac{4L_g + 5L_g\beta_1^2}{2(1-\beta_1)^2}\frac{1-\beta_1}{1-\beta_2}\frac{1}{1-\gamma}$$

$$=\frac{\sqrt{d}}{\sqrt{T}}D_f + \frac{1}{T}\mathbb{E}\left[\left(\frac{\eta G_{zo}}{1-\beta_1} + \eta^2\right)\left\|\frac{1}{\sqrt{\hat{\mathbf{v}}_0}}\right\|_1\right] + \frac{\sqrt{d}}{\sqrt{T}}\frac{4L_g + 5L_g\beta_1^2}{2(1-\beta_1)^2}\frac{1-\beta_1}{1-\beta_2}\frac{1}{1-\gamma} \quad (43)$$

Since $\hat{\mathbf{V}}_{0,ii}^{1/2} \geq c, \forall i \in [d]$. By Lemma A1, we have

$$\|\hat{\mathbf{V}}_t^{-1/4}(\nabla f_\mu(x) - \nabla f_\mu(x))\|^2 \leq \frac{\mu^2 d^2 L^2}{4c} \quad (44)$$

Then we can easily adapt (43) to

$$\mathbb{E}\left[\|\hat{\mathbf{V}}_t^{-1/4}\nabla f(x_R)\|^2\right] \leq\frac{\mu^2 d^2 L^2}{2c} + 2\frac{\sqrt{d}}{\sqrt{T}}D_f + 2\frac{1}{T}\mathbb{E}\left[\left(\frac{\eta G_{zo}}{1-\beta_1} + 2\eta2\right)\left\|\frac{1}{\sqrt{\hat{\mathbf{v}}_0}}\right\|_1\right]$$

$$+ \frac{\sqrt{d}}{\sqrt{T}}\frac{4L_g + 5L_g\beta_1^2}{(1-\beta_1)^2}\frac{1-\beta_1}{1-\beta_2}\frac{1}{1-\gamma}$$

Substituting into $\mu$ finishes the proof. $\qquad\square$

### 2.3 Proof of Proposition 3

Upon defining $G_{\text{zo},i} := \max_{t\in[T]}|\hat{g}_{t,i}|$

by [50, Lemma 2.2], for a vector $\mathbf{u}$ sampled from a unit sphere in $\mathbb{R}^d$, we have for any $i \in [d]$,

$$P[|u_i| \geq \sqrt{\xi/d}] \leq \exp\left((1 - \xi + \log\xi)/2\right). \quad (45)$$

Let $\xi = 4\log\frac{dT}{\delta}$, and by the assumption of $\max(d,T) \geq 3$ we have $1 + \log\xi \leq \xi/2$. Thus, we obtain from (45) that

$$P[|u_i| \geq \sqrt{\xi/d}] \leq \exp\left(-\xi/4\right) = \exp\left(-\log(dT/\delta)\right) = \delta/dT. \quad (46)$$

Recall that the ZO gradient estimate $\hat{\mathbf{g}}_t$ is given by the form

$$\hat{\nabla}f(\mathbf{x}) = (d/\mu)[f(\mathbf{x} + \mu\mathbf{u}) - f(\mathbf{x})]\mathbf{u}. \quad (47)$$

By Lipschitz of $f$ under **A2**, the $i$th coordinate of the ZO gradient estimate (47) is upper bounded by $dL_c|u_i|$. Since $\mathbf{u}$ is drawn uniformly randomly from a unit sphere, by (46) we have

$$P[dL_c|u_i| \geq L_c\sqrt{\xi d}] \leq \delta/dT. \quad (48)$$

Also, since $|\hat{g}_{t,i}| \leq dL_c|u_i|$, based on (48) we obtain that

$$P[|\hat{g}_{t,i}| \geq L_c\sqrt{\xi d}] \leq P[dL_c|u_i| \geq L_c\sqrt{\xi d}] \leq \delta/dT. \quad (49)$$

Substituting $\xi = 4\log\frac{dT}{\delta}$ into (49), we have

$$P[|\hat{g}_{t,i}| \geq 2L_c\sqrt{d\log(dT/\delta)}] \leq \delta/dT \quad (50)$$

Then by the union bound and (50), we have

$$P[|\hat{g}_{t,i}| \geq 2L_c\sqrt{d\log(dT/\delta)}, \forall i, t]$$

$$\leq \sum_{t\in[T]}\sum_{i\in[d]}P[|\hat{g}_{t,i}| \geq 2L_c\sqrt{d\log(dT/\delta)}] \leq dT(\delta/dT) = \delta,$$

which implies the inequality (11). $\qquad\square$

### 2.4 Proof of Theorem 1

The idea is to prove a similar result as Proposition 2 conditioned on the event in Proposition 3 $(\max_{t\in[T]}\{\|\hat{\mathbf{g}}_t\|_\infty\} \leq 2L_c\sqrt{d\log(dT/\delta)})$. Thus, the proof follows the same flow as Proposition 2. The difference is that (39) does not hold conditioned on the event and more efforts are need to bound

the corresponding term in (39). Denote the event that $\max_{t\in[T]}\{\|\hat{\mathbf{g}}_t\|_\infty\} \le 2L_c\sqrt{d\log(dT/\delta)}$ to be $U(\delta)$, we need to upper bound

$$\mathbb{E}\left[\langle\nabla f_\mu(\mathbf{x}_t), \alpha_{t-1}\hat{\mathbf{V}}_{t-1}^{-1/2}(\hat{\mathbf{g}}_t - \nabla f_\mu(\mathbf{x}_t))\rangle|U(\delta)\right] \tag{51}$$

where $\mathbb{E}[\cdot|U(\delta)]$ is conditional expectation conditioned on $U(\delta)$.

By Proposition 3, we know $P(U(\delta)) \ge 1 - \delta$ and using the fact that $E[\cdot|A] = \frac{E[\cdot] - E[\cdot|A^c]P(A^c)}{P(A)}$ for any event $A$ and its complimentary event $A^c$, we have

$$\mathbb{E}\left[\langle\nabla f_\mu(\mathbf{x}_t), \alpha_{t-1}\hat{\mathbf{V}}_{t-1}^{-1/2}(\hat{\mathbf{g}}_t - \nabla f_\mu(\mathbf{x}_t))\rangle|U(\delta)\right]$$

$$\le \frac{\mathbb{E}\left[\langle\nabla f_\mu(\mathbf{x}_t), \alpha_{t-1}\hat{\mathbf{V}}_{t-1}^{-1/2}(\hat{\mathbf{g}}_t - \nabla f_\mu(\mathbf{x}_t))\rangle\right]}{1-\delta}$$

$$+ \frac{\delta\left|\mathbb{E}\left[\langle\nabla f_\mu(\mathbf{x}_t), \alpha_{t-1}\hat{\mathbf{V}}_{t-1}^{-1/2}(\hat{\mathbf{g}}_t - \nabla f_\mu(\mathbf{x}_t))\rangle|U(\delta)^c\right]\right|}{1-\delta} \tag{52}$$

and further we have

$$\left|\mathbb{E}\left[\langle\nabla f_\mu(\mathbf{x}_t), \alpha_{t-1}\hat{\mathbf{V}}_{t-1}^{-1/2}(\hat{\mathbf{g}}_t - \nabla f_\mu(\mathbf{x}_t))\rangle|U(\delta)^c\right]\right|$$

$$\le d\frac{\alpha_{t-1}}{c}(\eta^2 + \eta\max_{t\in[T]}\|\hat{g}_t\|_\infty)$$

$$\le d\frac{\alpha_{t-1}}{c}(\eta^2 + \eta dL_c) \tag{53}$$

where the first inequality is due to $\|\nabla f_\mu(x_t)\|_\infty \le \eta$ and $\hat{v}_{t-1}^{1/2} \ge \hat{\mathbf{v}}_0^{1/2} \ge c\mathbf{1}$, the second inequality is due to (1) and Lipschitz continuity of $f(\mathbf{x};\boldsymbol{\xi})$.

Using the fact that $\mathbb{E}\left[\langle\nabla f_\mu(\mathbf{x}_t), \alpha_{t-1}\hat{\mathbf{V}}_{t-1}^{-1/2}(\hat{\mathbf{g}}_t - \nabla f_\mu(\mathbf{x}_t))\rangle\right] = 0$ proved in in (39) and set $\delta = 1/Td^{0.5}$, we have for $T \ge 2$

$$\mathbb{E}\left[\langle\nabla f_\mu(\mathbf{x}_t), \alpha_{t-1}\hat{\mathbf{V}}_{t-1}^{-1/2}(\hat{\mathbf{g}}_t - \nabla f_\mu(\mathbf{x}_t))\rangle|U(1/Td^{0.5})\right]$$

$$\le 2\frac{1}{Td^{0.5}}d\frac{\alpha_{t-1}}{c}(\eta^2 + \eta dL_c) = 2\frac{d^{1.5}}{T}\frac{\alpha_{t-1}}{c}\eta L_c + 2\frac{d^{0.5}}{T}\frac{\alpha_{t-1}}{c}\eta^2 \tag{54}$$

Replacing (39) with (54) and going through the rest of the proof of Proposition (2), one can finally get

$$\mathbb{E}\left[\|\hat{\mathbf{V}}_t^{-1/4}\nabla f(x_R)\|^2|U(1/Td^{0.5})\right]$$

$$\le \frac{\mu^2 d^2 L^2}{2c} + 2\frac{\sqrt{d}}{\sqrt{T}}D_f + 2\frac{1}{T}\mathbb{E}\left[\left(\frac{\eta G_{zo}}{1-\beta_1} + 2\eta 2\right)\left\|\frac{1}{\sqrt{\hat{\mathbf{v}}_0}}\right\|_1\bigg|U(1/Td^{0.5})\right]$$

$$+ \frac{\sqrt{d}}{\sqrt{T}}\frac{4L_g + 5L_g\beta_1^2}{(1-\beta_1)^2}\frac{1-\beta_1}{1-\beta_2}\frac{1}{1-\gamma} + 2\frac{d^{1.5}}{T}\frac{\eta L_c}{c} + 2\frac{d^{0.5}}{T}\frac{\eta^2}{c}.$$

Since in the event of $U(1/Td^{0.5})$, we have

$$G_{zo} = \max_{t\in[T]}\{\|\hat{\mathbf{g}}_t\|_\infty\} \le 2L_c\sqrt{d\log(d^{1.5}T^2)} = 2L_c\sqrt{d}\sqrt{1.5\log d + 2\log T}. \tag{55}$$

Substituting the above inequality into (55), we get the desired result. □

## 2.5 Proof of Theorem 2

To proceed into proof of Theorem 2, we give a few technical lemmas for the properties of (7).

**Lemma 2.5** *For any symmetric* $\mathbf{H} \succeq 0, \mathbf{g}, \omega$, *we have*

$$\langle\mathbf{g}, P_{\mathcal{X},\mathbf{H}}(\mathbf{x}^-, \mathbf{g}, \omega)\rangle \ge \|\mathbf{H}^{1/2}P_{\mathcal{X},\mathbf{H}}(\mathbf{x}^-, \mathbf{g}, \omega)\|^2 \tag{56}$$

**Proof of Lemma 2.5:** By definition of $\mathbf{x}^+$, the optimality condition of (6) is

$$\langle \mathbf{g} + \frac{1}{\omega}\mathbf{H}(\mathbf{x}^+ - \mathbf{x}^-), \mathbf{x} - \mathbf{x}^+ \rangle \geq 0 \quad \forall \mathbf{x} \in \mathcal{X}$$

Thus

$$\langle \mathbf{g} + \frac{1}{\omega}\mathbf{H}(\mathbf{x}^+ - x), \mathbf{x} - \mathbf{x}^+ \rangle \geq 0$$

which can be rearranged to

$$\langle \mathbf{g}, P_{\mathcal{X},\mathbf{H}}(\mathbf{x}^-, \mathbf{g}, \omega) \rangle = \frac{1}{\omega}\langle \mathbf{g}, x - \mathbf{x}^+ \rangle \geq \frac{1}{\omega^2}\langle \mathbf{H}(x - \mathbf{x}^+), x - \mathbf{x}^+ \rangle = \|\mathbf{H}^{1/2}P_{\mathcal{X},\mathbf{H}}(\mathbf{x}^-, \mathbf{g}, \omega)\|^2$$

This completes the proof. $\square$

**Lemma 2.6** *Let* $\mathbf{x}_1^+$ *and* $\mathbf{x}_2^+$ *be given by* (6) *with* $\mathbf{g}$ *replaced by* $\mathbf{g}_1$ *and* $\mathbf{g}_2$, *with* $H \succ 0$, *we have*

$$\|\mathbf{x}_1^+ - \mathbf{x}_2^+\| \leq \frac{\omega}{\lambda_{\min}(\mathbf{H})}\|\mathbf{g}_1 - \mathbf{g}_2\| \tag{57}$$

$$\|\mathbf{H}^{1/2}(\mathbf{x}_1^+ - \mathbf{x}_2^+)\| \leq \omega\|\mathbf{H}^{-1/2}(\mathbf{g}_1 - \mathbf{g}_2)\|. \tag{58}$$

*where* $\lambda_{\min}(\mathbf{H})$ *is the minimum eigenvalue of* $\mathbf{H}$.

**Proof of Lemma 2.6:** By definition of $\mathbf{x}^+$, the optimality condition of (6) is

$$\langle \mathbf{g} + \frac{1}{\omega}\mathbf{H}(\mathbf{x}^+ - \mathbf{x}^-), \mathbf{x} - \mathbf{x}^+ \rangle \geq 0 \quad \forall \mathbf{x} \in \mathcal{X}$$

Thus, we have

$$\langle \mathbf{g}_1 + \frac{1}{\omega}\mathbf{H}(\mathbf{x}_1^+ - \mathbf{x}^-), \mathbf{x}_2^+ - \mathbf{x}_1^+ \rangle \geq 0$$

$$\langle \mathbf{g}_2 + \frac{1}{\omega}\mathbf{H}(\mathbf{x}_2^+ - \mathbf{x}^-), \mathbf{x}_1^+ - \mathbf{x}_2^+ \rangle \geq 0$$

Summing up the above two inequalities, we get

$$\langle \mathbf{g}_1 - \mathbf{g}_2, \mathbf{x}_2^+ - \mathbf{x}_1^+ \rangle \geq \frac{1}{\omega}\langle \mathbf{H}(\mathbf{x}_2^+ - \mathbf{x}_1^+), \mathbf{x}_2^+ - \mathbf{x}_1^+ \rangle \tag{59}$$

By Cauchy-Schwarz inequality, we get

$$\|\mathbf{g}_1 - \mathbf{g}_2\|\|\mathbf{x}_2^+ - \mathbf{x}_1^+\| \geq \langle \mathbf{g}_1 - \mathbf{g}_2, \mathbf{x}_2^+ - \mathbf{x}_1^+ \rangle \geq \frac{1}{\omega}\langle \mathbf{H}(\mathbf{x}_2^+ - \mathbf{x}_1^+), \mathbf{x}_2^+ - \mathbf{x}_1^+ \rangle$$

$$\geq \frac{1}{\omega}\lambda_{\min}(\mathbf{H})\|\mathbf{x}_2^+ - \mathbf{x}_1^+\|^2$$

which gives (57).

Further, by (59) and Cauchy-Schwarz, we also have

$$\|\mathbf{H}^{-1/2}(\mathbf{g}_1 - \mathbf{g}_2)\|\|\mathbf{H}^{1/2}(\mathbf{x}_2^+ - \mathbf{x}_1^+)\| \geq \langle \mathbf{g}_1 - \mathbf{g}_2, \mathbf{x}_2^+ - \mathbf{x}_1^+ \rangle$$

$$\geq \frac{1}{\omega}\langle \mathbf{H}(\mathbf{x}_2^+ - \mathbf{x}_1^+), \mathbf{x}_2^+ - \mathbf{x}_1^+ \rangle = \frac{1}{\omega}\|\mathbf{H}^{1/2}(\mathbf{x}_2^+ - \mathbf{x}_1^+)\|^2$$

which gives (58). This completes the proof. $\square$

The following lemma characterizes the difference between projected points if different distance matrices are used in ZO-AdaMM.

**Lemma 2.7** *Assume* $\mathbf{V}_t^{1/2} \geq c\mathbf{I}$, *ZO-AdaMM yields*

$$\left\|\left(P_{\mathcal{X},\hat{\mathbf{V}}_{t-1}^{1/2}}(\mathbf{x}_t, \nabla f_\mu(\mathbf{x}_t), \alpha_t) - P_{\mathcal{X},\hat{\mathbf{V}}_t^{1/2}}(\mathbf{x}_t, \nabla f_\mu(\mathbf{x}_t), \alpha_t)\right)\right\|^2 \leq \sum_{i=1}^d v_{t,i}^{1/2}(\hat{v}_{t,i}^{1/2} - \hat{v}_{t-1,i}^{1/2})\frac{1}{c^4}\eta^2. \tag{60}$$

**Proof of Lemma 2.7:** Recall the optimality condition of (6) is

$$\langle \mathbf{g} + \frac{1}{\omega}\mathbf{H}(\mathbf{x}^+ - \mathbf{x}^-), \mathbf{x} - \mathbf{x}^+ \rangle \geq 0 \quad \forall \mathbf{x} \in \mathcal{X} \tag{61}$$

Let us define

$$\mathbf{x}_t^* \triangleq \mathbf{x}_t - \alpha_t P_{\mathcal{X}, \hat{\mathbf{V}}_{t-1}^{1/2}}(\mathbf{x}_t, \nabla f_\mu(\mathbf{x}_t), \alpha_t)$$

$$\tilde{\mathbf{x}}_t^* \triangleq \mathbf{x}_t - \alpha_t P_{\mathcal{X}, \hat{\mathbf{V}}_t^{1/2}}(\mathbf{x}_t, \nabla f_\mu(\mathbf{x}_t), \alpha_t).$$

By optimality condition (61), we have

$$\langle \nabla f_\mu(\mathbf{x}_t) + \frac{1}{\alpha_t}\hat{\mathbf{V}}_t^{1/2}(\tilde{\mathbf{x}}_t^* - \mathbf{x}_t), \mathbf{x}_t^* - \tilde{\mathbf{x}}_t^* \rangle \geq 0$$

$$\langle \nabla f_\mu(\mathbf{x}_t) + \frac{1}{\alpha_t}\hat{\mathbf{V}}_{t-1}^{1/2}(\mathbf{x}_t^* - \mathbf{x}_t), \tilde{\mathbf{x}}_t^* - \mathbf{x}_t^* \rangle \geq 0$$

Summing the above up, we get

$$\langle \hat{\mathbf{V}}_t^{1/2}(\tilde{\mathbf{x}}_t^* - \mathbf{x}_t) - \hat{\mathbf{V}}_{t-1}^{1/2}(\mathbf{x}_t^* - \mathbf{x}_t), \mathbf{x}_t^* - \tilde{\mathbf{x}}_t^* \rangle \geq 0$$

which is equivalent to

$$\langle (\hat{\mathbf{V}}_t^{1/2} - \hat{\mathbf{V}}_{t-1}^{1/2})(\mathbf{x}_t^* - \mathbf{x}_t), \mathbf{x}_t^* - \tilde{\mathbf{x}}_t^* \rangle$$
$$+ \langle \hat{\mathbf{V}}_t^{1/2}(\tilde{\mathbf{x}}_t^* - \mathbf{x}_t^*), \mathbf{x}_t^* - \tilde{\mathbf{x}}_t^* \rangle \geq 0.$$

Further rearranging, we have

$$\langle (\hat{\mathbf{V}}_t^{1/2} - \hat{\mathbf{V}}_{t-1}^{1/2})(\mathbf{x}_t^* - \mathbf{x}_t), \mathbf{x}_t^* - \tilde{\mathbf{x}}_t^* \rangle \geq \|\hat{\mathbf{V}}_t^{1/4}(\tilde{\mathbf{x}}_t^* - \mathbf{x}_t^*)\|^2 \geq c\|(\tilde{\mathbf{x}}_t^* - \mathbf{x}_t^*)\|^2$$

which implies (by using Cauchy-Swartz on the left hand side and then squaring both sides)

$$c^2\|(\tilde{\mathbf{x}}_t^* - \mathbf{x}_t^*)\|^2 \leq \|(\hat{\mathbf{V}}_t^{1/2} - \hat{\mathbf{V}}_{t-1}^{1/2})(\mathbf{x}_t^* - \mathbf{x}_t)\|^2 = \sum_{i=1}^d (\hat{v}_{t,i}^{1/2} - \hat{v}_{t-1,i}^{1/2})^2(\hat{x}_{t,i}^* - x_{t,i})^2$$

$$\overset{(a)}{\leq} \sum_{i=1}^d \hat{v}_{t,i}^{1/2}(\hat{v}_{t,i}^{1/2} - \hat{v}_{t-1,i}^{1/2})\|\hat{x}_t^* - x_t\|^2 \overset{(b)}{\leq} \sum_{i=1}^d \hat{v}_{t,i}^{1/2}(\hat{v}_{t,i}^{1/2} - \hat{v}_{t-1,i}^{1/2})\frac{1}{c^2}\alpha_t^2\|\nabla f_\mu(x_t)\|^2$$

$$\leq \sum_{i=1}^d \hat{v}_{t,i}^{1/2}(\hat{v}_{t,i}^{1/2} - \hat{v}_{t-1,i}^{1/2})\frac{1}{c^2}\alpha_t^2\eta^2 \tag{62}$$

where (a) is due to $\hat{v}_{t,i}^{1/2} \geq \hat{v}_{t-1,i}^{1/2}$ and (b) is due to Lemma 2.6 by treating $\mathbf{g}_1 = \nabla f_\mu(\mathbf{x}_t)$, $\mathbf{g}_2 = 0$, $\mathbf{x}^- = \mathbf{x}_t$, $H = \hat{\mathbf{V}}_t^{1/2}$. Substituting (7) into LHS of the above inequality and rearrange, we get (60). This completes the proof. $\square$

Now we are ready to prove our main theorem.

**Proof of Theorem 2:**

We start with standard decent lemma in nonconvex optimization. By Lipschitz smoothness of $f_\mu$, we have

$$f_\mu(\mathbf{x}_{t+1}) \leq f_\mu(\mathbf{x}_t) - \alpha_t\langle \nabla f_\mu(\mathbf{x}_t), P_{\mathcal{X}, \hat{\mathbf{V}}_t^{1/2}}(\mathbf{x}_t, \hat{\mathbf{g}}_t, \alpha_t)\rangle + \frac{L}{2}\alpha_t^2\|P_{\mathcal{X}, \hat{\mathbf{V}}_t^{1/2}}(\mathbf{x}_t, \hat{\mathbf{g}}_t, \alpha_t)\|^2. \tag{63}$$

We need to upper bound RHS of the above inequality and split out a descent quantity.

$$- \langle \nabla f_\mu(\mathbf{x}_t), P_{\mathcal{X}, \hat{\mathbf{V}}_t^{1/2}}(\mathbf{x}_t, \hat{\mathbf{g}}_t, \alpha_t)\rangle$$
$$= - \langle \hat{\mathbf{g}}_t, P_{\mathcal{X}, \hat{\mathbf{V}}_t^{1/2}}(\mathbf{x}_t, \hat{\mathbf{g}}_t, \alpha_t)\rangle + \langle \hat{\mathbf{g}}_t - \nabla f_\mu(\mathbf{x}_t), P_{\mathcal{X}, \hat{\mathbf{V}}_t^{1/2}}(\mathbf{x}_t, \hat{\mathbf{g}}_t, \alpha_t)\rangle$$
$$\leq - \|\hat{\mathbf{V}}_t^{1/4} P_{\mathcal{X}, \hat{\mathbf{V}}_t^{1/2}}(\mathbf{x}_t, \hat{\mathbf{g}}_t, \alpha_t)\|^2 + \langle \hat{\mathbf{g}}_t - \nabla f_\mu(\mathbf{x}_t), P_{\mathcal{X}, \hat{\mathbf{V}}_t^{1/2}}(\mathbf{x}_t, \hat{\mathbf{g}}_t, \alpha_t)\rangle \tag{64}$$

where the inequality is by Lemma (2.5) and some simple substitutions.

Further, for the last term in RHS of (64) we have

$$\langle \hat{\mathbf{g}}_t - \nabla f_\mu(\mathbf{x}_t), P_{\mathcal{X}, \hat{\mathbf{V}}_t^{1/2}}(\mathbf{x}_t, \hat{\mathbf{g}}_t, \alpha_t) \rangle$$

$$= \left. \begin{array}{l} + \langle \hat{\mathbf{g}}_t - \nabla f_\mu(\mathbf{x}_t), P_{\mathcal{X}, \hat{\mathbf{V}}_t^{1/2}}(\mathbf{x}_t, \hat{\mathbf{g}}_t, \alpha_t) \rangle \\ - \langle \hat{\mathbf{g}}_t - \nabla f_\mu(\mathbf{x}_t), P_{\mathcal{X}, \hat{\mathbf{V}}_t^{1/2}}(\mathbf{x}_t, \nabla f_\mu(\mathbf{x}_t), \alpha_t) \rangle \end{array} \right\} A$$

$$\left. \begin{array}{l} + \langle \hat{\mathbf{g}}_t - \nabla f_\mu(\mathbf{x}_t), P_{\mathcal{X}, \hat{\mathbf{V}}_t^{1/2}}(\mathbf{x}_t, \nabla f_\mu(\mathbf{x}_t), \alpha_t) \rangle \\ - \langle \hat{\mathbf{g}}_t - \nabla f_\mu(\mathbf{x}_t), P_{\mathcal{X}, \hat{\mathbf{V}}_{t-1}^{1/2}}(\mathbf{x}_t, \nabla f_\mu(\mathbf{x}_t), \alpha_t) \rangle \end{array} \right\} B$$

$$+ \underbrace{\langle \hat{\mathbf{g}}_t - \nabla f_\mu(\mathbf{x}_t), P_{\mathcal{X}, \hat{\mathbf{V}}_{t-1}^{1/2}}(\mathbf{x}_t, \nabla f_\mu(\mathbf{x}_t), \alpha_t) \rangle}_{C} \qquad (65)$$

Next, we bound the three terms in RHS of (65).

Let's bound term $A$ first, with the assumption $\hat{\mathbf{V}}^{1/2} \geq c\mathbf{I}$, by Lemma 2.6, (7) and Cauchy-Schwartz inequality, we have:

$$A = \langle \hat{\mathbf{g}}_t - \nabla f_\mu(\mathbf{x}_t), P_{\mathcal{X}, \hat{\mathbf{V}}_t^{1/2}}(\mathbf{x}_t, \hat{\mathbf{g}}_t, \alpha_t) - P_{\mathcal{X}, \hat{\mathbf{V}}_t^{1/2}}(\mathbf{x}_t, \nabla f_\mu(\mathbf{x}_t), \alpha_t) \rangle \leq \frac{1}{c} \|\hat{\mathbf{g}}_t - f_\mu(\mathbf{x}_t)\|^2 \quad (66)$$

Now let's bound term $C$, because $\mathbb{E}[\hat{\mathbf{g}}_t] = \nabla f_\mu(\mathbf{x}_t)$ and the noise in $\hat{\mathbf{g}}_t$ is independent of $\nabla f_\mu(\mathbf{x}_t)$ and $\hat{\mathbf{V}}_{t-1}$, we have

$$\mathbb{E}[\langle \nabla f_\mu(\mathbf{x}_t) - \hat{\mathbf{g}}_t, P_{\mathcal{X}, \hat{\mathbf{V}}_{t-1}^{1/2}}(\mathbf{x}_t, \nabla f_\mu(\mathbf{x}_t), \alpha_t) \rangle] = 0 \qquad (67)$$

Substituting the above bounds for A and C, into (65) and (64), using Young's inequality on term B, we have

$$- \mathbb{E}[\langle \nabla f_\mu(\mathbf{x}_t), P_{\mathcal{X}, \hat{\mathbf{V}}_t^{1/2}}(\mathbf{x}_t, \hat{\mathbf{g}}_t, \alpha_t) \rangle]$$

$$\leq - \mathbb{E}[\|\hat{\mathbf{V}}_t^{1/4} P_{\mathcal{X}, \hat{\mathbf{V}}_t^{1/2}}(\mathbf{x}_t, \hat{\mathbf{g}}_t, \alpha_t)\|^2] + \frac{1}{c}\mathbb{E}[\|\hat{\mathbf{g}}_t - f_\mu(\mathbf{x}_t)\|^2] + \frac{1}{2}\mathbb{E}[\|\hat{\mathbf{g}}_t - f_\mu(\mathbf{x}_t)\|^2] + \frac{1}{2}\mathbb{E}[B_2]$$
$$\qquad (68)$$

where we define

$$B_2 := \left\| (P_{\mathcal{X}, \hat{\mathbf{V}}_{t-1}^{1/2}}(\mathbf{x}_t, \nabla f_\mu(\mathbf{x}_t), \alpha_t) - P_{\mathcal{X}, \hat{\mathbf{V}}_t^{1/2}}(\mathbf{x}_t, \nabla f_\mu(\mathbf{x}_t), \alpha_t)) \right\|^2 .$$

What remains is to bound the term $B2$ which is given by Lemma 2.7.

Combining (63), (68), (60), we have

$$\mathbb{E}[f_\mu(\mathbf{x}_{t+1})] \leq \mathbb{E}[f_\mu(\mathbf{x}_t)] - \alpha_t \mathbb{E}[\|\hat{\mathbf{V}}_t^{1/4} P_{\mathcal{X}, \hat{\mathbf{V}}_t^{1/2}}(\mathbf{x}_t, \hat{\mathbf{g}}_t, \alpha_t)\|^2] + \alpha_t(\frac{1}{c} + \frac{1}{2})\mathbb{E}[\|\hat{\mathbf{g}}_t - f_\mu(\mathbf{x}_t)\|^2]$$

$$+ \alpha_t \frac{1}{2}\mathbb{E}\Big[\sum_{i=1}^{d} \hat{v}_{t,i}^{1/2}(\hat{v}_{t,i}^{1/2} - \hat{v}_{t-1,i}^{1/2})\frac{1}{c^4}\eta^2\Big] + \frac{L}{2}\alpha_t^2 \mathbb{E}\left[\frac{1}{c^2}\|\hat{\mathbf{V}}_t^{1/4} P_{\mathcal{X}, \hat{\mathbf{V}}_t^{1/2}}(\mathbf{x}_t, \hat{\mathbf{g}}_t, \alpha_t)\|^2\right]$$
$$\qquad (69)$$

which can be rearranged into

$$(\alpha_t - \frac{L}{2c^2}\alpha_t^2)\mathbb{E}[\|\hat{\mathbf{V}}_t^{1/4} P_{\mathcal{X}, \hat{\mathbf{V}}_t^{1/2}}(\mathbf{x}_t, \hat{\mathbf{g}}_t, \alpha_t)\|^2]$$

$$\leq \mathbb{E}[f_\mu(\mathbf{x}_t)] - \mathbb{E}[f_\mu(\mathbf{x}_{t+1})] + \alpha_t(\frac{1}{c} + \frac{1}{2})\mathbb{E}[\|\hat{\mathbf{g}}_t - f_\mu(\mathbf{x}_t)\|^2]$$

$$+ \alpha_t \frac{1}{2}\mathbb{E}\left[\sum_{i=1}^{d} \hat{v}_{t,i}^{1/2}(\hat{v}_{t,i}^{1/2} - \hat{v}_{t-1,i}^{1/2})\frac{1}{c^4}\eta^2\right]. \qquad (70)$$

In addition, we have

$$\|\hat{\mathbf{V}}_t^{1/4} P_{\mathcal{X},\hat{\mathbf{V}}_t^{1/2}}(\mathbf{x}_t, \nabla f(\mathbf{x}_t), \alpha_t)\|^2 \leq 3\|\hat{\mathbf{V}}_t^{1/4} P_{\mathcal{X},\hat{\mathbf{V}}_t^{1/2}}(\mathbf{x}_t, \hat{\mathbf{g}}_t, \alpha_t)\|^2$$

$$+ 3\|\hat{\mathbf{V}}_t^{1/4}(P_{\mathcal{X},\hat{\mathbf{V}}_t^{1/2}}(\mathbf{x}_t, \nabla f_\mu(\mathbf{x}_t), \alpha_t) - P_{\mathcal{X},\hat{\mathbf{V}}_t^{1/2}}(\mathbf{x}_t, \nabla f(\mathbf{x}_t), \alpha_t))\|^2$$

$$+ 3\|\hat{\mathbf{V}}_t^{1/4}(P_{\mathcal{X},\hat{\mathbf{V}}_t^{1/2}}(\mathbf{x}_t, \hat{\mathbf{g}}_t, \alpha_t) - P_{\mathcal{X},\hat{\mathbf{V}}_t^{1/2}}(\mathbf{x}_t, \nabla f_\mu(\mathbf{x}_t), \alpha_t))\|^2$$

$$\leq 3\|\hat{\mathbf{V}}_t^{1/4} P_{\mathcal{X},\hat{\mathbf{V}}_t^{1/2}}(\mathbf{x}_t, \hat{\mathbf{g}}_t, \alpha_t)\|^2 + \frac{3}{c}\|\nabla f_\mu(\mathbf{x}_t) - \nabla f(\mathbf{x}_t)\|^2 + \frac{3}{c}\|\hat{\mathbf{g}}_t - \nabla f_\mu(\mathbf{x}_t)\|^2 \qquad (71)$$

where the second inequality is by (7) and Lemma (2.6)

Combining (71) and (70), we have

$$\left(\alpha_t - \frac{L}{2c^2}\alpha_t^2\right)\|\hat{\mathbf{V}}_t^{1/4} P_{\mathcal{X},\hat{\mathbf{V}}_t^{1/2}}(\mathbf{x}_t, \nabla f(\mathbf{x}_t), \alpha_t)\|^2$$

$$\leq 3(\mathbb{E}[f_\mu(\mathbf{x}_t)] - \mathbb{E}[f_\mu(\mathbf{x}_{t+1})]) + (3\alpha_t(\frac{1}{c} + \frac{1}{2}) + \frac{3}{c}(\alpha_t - \frac{L}{2c^2}\alpha_t^2))\mathbb{E}[\|\hat{\mathbf{g}}_t - f_\mu(\mathbf{x}_t)\|^2]$$

$$+ \frac{3}{2}\alpha_t\mathbb{E}[\sum_{i=1}^d \hat{v}_{t,i}^{1/2}(\hat{v}_{t,i}^{1/2} - \hat{v}_{t-1,i}^{1/2})\frac{1}{c^4}\eta^2] + \frac{3}{c}(\alpha_t - \frac{L}{2c^2}\alpha_t^2)\|\nabla f_\mu(\mathbf{x}_t) - \nabla f(\mathbf{x}_t)\|^2 \qquad (72)$$

Summing over $t$ from 1 to $T$, setting $\alpha_t = \alpha$, and dividing both sides by $T(\alpha - \frac{L_g\alpha^2}{2c^2})$, we get

$$\frac{1}{T}\sum_{t=1}^T \mathbb{E}[\|\hat{\mathbf{V}}_t^{1/4} P_{\mathcal{X},\hat{\mathbf{V}}_t^{1/2}}(\mathbf{x}_t \nabla f(\mathbf{x}_t), \alpha_t)\|^2]$$

$$\leq \frac{3}{T(\alpha - \frac{L_g\alpha^2}{2c})}(\mathbb{E}[f_\mu(\mathbf{x}_1)] - \mathbb{E}[f_\mu(x_{T+1})]) + \left(\frac{3\alpha(c+2)}{2Tc(\alpha - \frac{L_g\alpha^2}{2c})} + \frac{3}{Tc}\right)\sum_{t=1}^T \mathbb{E}[\|\hat{\mathbf{g}}_t - f_\mu(\mathbf{x}_t)\|^2]$$

$$+ \frac{3\alpha}{2T(\alpha - \frac{L_g\alpha^2}{2c})}\mathbb{E}[\sum_{i=1}^d \hat{v}_{T,i}]\frac{1}{c^4}\eta^2 + \frac{3}{Tc}\sum_{t=1}^T \mathbb{E}[\|\nabla f_\mu(\mathbf{x}_t) - \nabla f(\mathbf{x}_t)\|^2]. \qquad (73)$$

Choose $\alpha \leq \frac{c}{L}$, we have

$$\alpha - \frac{L_g\alpha^2}{2c} = \alpha\left(1 - \frac{L_g\alpha}{2c}\right) \geq \alpha(1 - \frac{1}{2}) = \frac{\alpha}{2} \qquad (74)$$

and (73) becomes

$$\frac{1}{T}\sum_{t=1}^T \mathbb{E}\left[\|\hat{\mathbf{V}}_t^{1/4} P_{\mathcal{X},\hat{\mathbf{V}}_t^{1/2}}(\mathbf{x}_t \nabla f(\mathbf{x}_t), \alpha_t)\|^2\right]$$

$$\leq \frac{6}{T\alpha}D_f + \frac{1}{T}(\frac{9}{c} + 3)\sum_{t=1}^T \mathbb{E}\left[\|\hat{\mathbf{g}}_t - f_\mu(\mathbf{x}_t)\|^2\right] + \frac{3}{T}\frac{1}{c^4}\eta^2\mathbb{E}\left[\sum_{i=1}^d \hat{v}_{T,i}\right] + \frac{3}{c}\frac{\mu^2 d^2 L_g^2}{4} \qquad (75)$$

where we defined $D_f := \mathbb{E}[f_\mu(\mathbf{x}_1)] - \min_x f_\mu(x)$ and used the fact that $\|\nabla f_\mu(\mathbf{x}_t) - \nabla f(\mathbf{x}_t)\|^2 \leq \frac{\mu^2 d^2 L_g^2}{4}$ by Lemma A1.

Further, we have

$$\mathbb{E}\left[\sum_{i=1}^d \hat{v}_{T,i}\right] = \mathbb{E}\left[\sum_{i=1}^d \max_{t\in[T]}(1-\beta_2)\sum_{k=1}^t \beta_2^{t-k}\hat{g}_{k,i}^2\right]$$

$$\leq \mathbb{E}\left[d\max_{t\in[T]}(1-\beta_2)\sum_{k=1}^t \beta_2^{t-k}\|\hat{g}_k\|_\infty\right]$$

$$\leq \mathbb{E}\left[d\max_{t\in[T]}\|\hat{g}_t\|_\infty\right]$$

$$(76)$$

where the last inequality holds since $\sum_{k=1}^T \beta_2^{T-k} \leq 1/(1-\beta_2)$.

Uniformly randomly picking $R$ from 1 to $T$ and substituting (76) into (75) finishes the proof. $\qquad \square$

# 3 Proof for Convex Optimization

## 3.1 Proof of Proposition 4

We follow the analytic framework in [18, Theorem 4] Based on Lemma A1, we obtain that $f_{t,\mu}$ defined in (2) (with respect to $f_t$) is convex. The convexity of $f_{t,\mu}$ yields

$$f_{t,\mu}(\mathbf{x}_t) - f_{t,\mu}(\mathbf{x}^*) \leq \langle \mathbb{E}_\mathbf{u}[\hat{\mathbf{g}}_t], \mathbf{x}_t - \mathbf{x}^* \rangle, \tag{77}$$

where we have used the fact that $\mathbb{E}_\mathbf{u}[\hat{\mathbf{g}}_t] = \nabla f_{t,\mu}(\mathbf{x}_t)$ given by Lemma A1. Taking the expectation with respect to all the randomness in (77), we then obtain

$$\mathbb{E}[f_{t,\mu}(\mathbf{x}_t) - f_{t,\mu}(\mathbf{x}^*)] \leq \mathbb{E}\langle \hat{\mathbf{g}}_t, \mathbf{x}_t - \mathbf{x}^* \rangle. \tag{78}$$

Further, recall that $\Pi_{\mathcal{X}, \sqrt{\hat{\mathbf{V}}_t}}(\mathbf{x}^*) = \arg\min_{\mathbf{x} \in \mathcal{X}} \|\hat{\mathbf{V}}_t^{1/4}(\mathbf{x} - \mathbf{x}^*)\|^2 = \mathbf{x}^*$, where for ease of notation, let $\|\cdot\|$ denote the Euclidean norm. Applying [18, Lemma 4] to ZO-AdaMM, we obtain that

$$\left\| \hat{\mathbf{V}}_t^{1/4}(\mathbf{x}_{t+1} - \mathbf{x}^*) \right\|^2 \leq \left\| \hat{\mathbf{V}}_t^{1/4}(\mathbf{x}_t - \alpha_t \hat{\mathbf{V}}_t^{-1/2} \mathbf{m}_t - \mathbf{x}^*) \right\|^2$$

$$= \left\| \hat{\mathbf{V}}_t^{1/4}(\mathbf{x}_t - \mathbf{x}^*) \right\|^2 + \alpha_t^2 \|\hat{\mathbf{V}}_t^{-1/4} \mathbf{m}_t\|^2 - 2\alpha_t \langle \beta_{1,t} \mathbf{m}_{t-1} + (1 - \beta_{1,t})\hat{\mathbf{g}}_t, \mathbf{x}_t - \mathbf{x}^* \rangle. \tag{79}$$

Rearranging the above inequality, and using the Cauchy-Schwarz inequality $2\langle \mathbf{a}, \mathbf{b} \rangle \leq c\|\mathbf{a}\|^2 + \frac{1}{c}\|\mathbf{b}\|^2$ for $c > 0$, we obtain

$$\langle \hat{\mathbf{g}}_t, \mathbf{x}_t - \mathbf{x}^* \rangle \leq \frac{\|\hat{\mathbf{V}}_t^{1/4}(\mathbf{x}_t - \mathbf{x}^*)\|^2 - \|\hat{\mathbf{V}}_t^{1/4}(\mathbf{x}_{t+1} - \mathbf{x}^*)\|^2}{2\alpha_t(1 - \beta_{1,t})} + \frac{\alpha_t \|\hat{\mathbf{V}}_t^{-1/4} \mathbf{m}_t\|^2}{2(1 - \beta_{1,t})}$$

$$+ \frac{\beta_{1,t}}{1 - \beta_{1,t}} \frac{\alpha_t \|\hat{\mathbf{V}}_t^{-1/4} \mathbf{m}_{t-1}\|^2}{2} + \frac{\beta_{1,t}}{1 - \beta_{1,t}} \frac{\|\hat{\mathbf{V}}_t^{1/4}(\mathbf{x}_t - \mathbf{x}^*)\|^2}{2\alpha_t}. \tag{80}$$

Taking the sum over $t$ for (80), we obtain

$$\mathbb{E}\left[ \sum_{t=1}^T \langle \hat{\mathbf{g}}_t, \mathbf{x}_t - \mathbf{x}^* \rangle \right] \leq \frac{1}{2(1 - \beta_1)} \mathbb{E}\underbrace{\left[ \sum_{t=1}^T \alpha_t \|\hat{\mathbf{V}}_t^{-1/4} \mathbf{m}_t\|^2 \right]}_{A} + \frac{\beta_1}{2(1 - \beta_1)} \mathbb{E}\underbrace{\left[ \sum_{t=1}^T \alpha_t \|\hat{\mathbf{V}}_t^{-1/4} \mathbf{m}_{t-1}\|^2 \right]}_{B}$$

$$+ \sum_{t=1}^T \mathbb{E}\left[ \frac{\|\hat{\mathbf{V}}_t^{1/4}(\mathbf{x}_t - \mathbf{x}^*)\|^2 - \|\hat{\mathbf{V}}_t^{1/4}(\mathbf{x}_{t+1} - \mathbf{x}^*)\|^2}{2\alpha_t(1 - \beta_{1,t})} \right] + \sum_{t=1}^T \mathbb{E}\left[ \frac{\beta_{1,t}}{2\alpha_t(1 - \beta_1)} \|\hat{\mathbf{V}}_t^{1/4}(\mathbf{x}_t - \mathbf{x}^*)\|^2 \right], \tag{81}$$

where we have used the facts that $\beta_{1,t} \leq \beta_1$ and $1/(1 - \beta_{1,t}) \leq 1/(1 - \beta_1)$.

We next bound term $A$ in (81). Based on (4), we can directly apply [18, Lemma 2] to obtain that

$$A \leq \frac{\alpha\sqrt{1 + \log T}}{(1 - \beta_1)(1 - \gamma)\sqrt{1 - \beta_2}} \sum_{i=1}^d \|\hat{\mathbf{g}}_{1:T,i}\|_2. \tag{82}$$

Furthermore, we bound term $B$ in (81). Based on (4), we obtain that

$$B = \sum_{t=1}^{T-1} \alpha_t \|\hat{\mathbf{V}}_t^{-1/4} \mathbf{m}_{t-1}\|^2 + \alpha_T \sum_{i=1}^d \frac{m_{T-1,i}^2}{\sqrt{\hat{v}_{T,i}}}$$

$$\leq \sum_{t=1}^{T-1} \alpha_t \|\hat{\mathbf{V}}_t^{-1/4} \mathbf{m}_{t-1}\|^2 + \alpha_T \sum_{i=1}^d \frac{m_{T-1,i}^2}{\sqrt{v_{T,i}}}, \tag{83}$$

where we have used the fact that $\mathbf{v}_t \leq \hat{\mathbf{v}}_t$ given in Algorithm 1. The last term in (83) can be further derived via (4),

$$\alpha_T \sum_{i=1}^d \frac{m_{T-1,i}^2}{\sqrt{v_{T,i}}} = \alpha \sum_{i=1}^d \frac{\left(\sum_{j=1}^{T-1} \left[\left(\prod_{k=1}^{T-j-1} \beta_{1,T-k}\right) \hat{g}_{j,i}(1 - \beta_{1,j})\right]\right)^2}{\sqrt{T(1-\beta_2)\sum_{j=1}^T (\beta_2^{T-j}\hat{g}_{j,i}^2)}}$$

$$\leq \alpha \sum_{i=1}^d \frac{\left(\sum_{j=1}^{T-1} \beta_1^{T-1-j}(1 - \beta_{1,j})^2\right)\left(\sum_{j=1}^{T-1} \beta_1^{T-1-j}\hat{g}_{j,i}^2\right)}{\sqrt{T(1-\beta_2)\sum_{j=1}^T (\beta_2^{T-j}\hat{g}_{j,i}^2)}}$$

$$\leq \alpha \sum_{i=1}^d \frac{\left(\sum_{j=1}^{T} \beta_1^{T-1-j}\right)\left(\sum_{j=1}^{T-1} \beta_1^{T-1-j}\hat{g}_{j,i}^2\right)}{\sqrt{T(1-\beta_2)\sum_{j=1}^T (\beta_2^{T-j}\hat{g}_{j,i}^2)}}$$

$$\leq \frac{\alpha}{(1-\beta_1)\sqrt{T(1-\beta_2)}} \sum_{i=1}^d \sum_{j=1}^T \frac{\beta_1^{T-1-j}\hat{g}_{j,i}^2}{\sqrt{\beta_2^{T-j}\hat{g}_{j,i}^2}}$$

$$= \frac{\alpha}{\beta_1(1-\beta_1)\sqrt{T(1-\beta_2)}} \sum_{i=1}^d \sum_{j=1}^T \gamma^{T-j}|\hat{g}_{j,i}|, \tag{84}$$

where the first inequality holds due to Cauchy-Schwarz inequality and $\beta_{1,T-k} \leq \beta_1$ for $\forall k$, the second inequality holds due to $1 - \beta_{1,j} \leq 1$, and the third inequality holds due to $\sum_{j=1}^T \beta_1^{T-1-j} \leq 1/(1-\beta_1)$ and $\beta_2^{T-j}\hat{g}_{j,i}^2 \leq \sum_{j=1}^T \beta_2^{T-j}\hat{g}_{j,i}^2$. Based on (84), we then applies the proof of [18, Lemma 2], which yields

$$B \leq \frac{\alpha\sqrt{1+\log T}}{\beta_1(1-\beta_1)(1-\gamma)\sqrt{1-\beta_2}} \sum_{i=1}^d \|\hat{\mathbf{g}}_{1:T,i}\|_2 \tag{85}$$

Substituting (82) and (85) into (81), we obtain that

$$\mathbb{E}\left[\sum_{t=1}^T \langle \hat{\mathbf{g}}_t, \mathbf{x}_t - \mathbf{x}^* \rangle\right] \leq \frac{\alpha\sqrt{1+\log T}\sum_{i=1}^d \mathbb{E}\|\hat{\mathbf{g}}_{1:T,i}\|}{(1-\beta_1)^2(1-\gamma)\sqrt{1-\beta_2}}$$

$$+ \mathbb{E}\underbrace{\left[\sum_{t=1}^T \frac{\|\hat{\mathbf{V}}_t^{1/4}(\mathbf{x}_t - \mathbf{x}^*)\|^2 - \|\hat{\mathbf{V}}_t^{1/4}(\mathbf{x}_{t+1} - \mathbf{x}^*)\|^2}{2\alpha_t(1-\beta_{1,t})}\right]}_{C} + \mathbb{E}\underbrace{\left[\sum_{t=1}^T \frac{\beta_{1,t}\|\hat{\mathbf{V}}_t^{1/4}(\mathbf{x}_t - \mathbf{x}^*)\|^2}{2\alpha_t(1-\beta_1)}\right]}_{D}. \tag{86}$$

In (86), the term $D$ yields

$$D \leq \frac{\beta_1 D_\infty^2}{2(1-\beta_1)} \sum_{t=1}^T \sum_{i=1}^d \frac{\hat{v}_{t,i}^{1/2}}{\alpha_t}. \tag{87}$$

*We remark that it was shown in [39] that the proof in [18] to bound the term $C$ is problematic. Compared to [39], we propose a simpler fix to bound $C$ when $0 < \beta_{1,t} \leq \beta_{1,t-1} \leq 1$. We rewrite $C$ in (86) as*

$$C = \frac{\|\hat{\mathbf{V}}_1^{1/4}(\mathbf{x}_1 - \mathbf{x}^*)\|^2}{2\alpha_1(1-\beta_{1,1})} + \sum_{t=2}^T \frac{\|\hat{\mathbf{V}}_t^{1/4}(\mathbf{x}_t - \mathbf{x}^*)\|^2}{2\alpha_t(1-\beta_{1,t})}$$

$$- \sum_{t=2}^T \frac{\|\hat{\mathbf{V}}_{t-1}^{1/4}(\mathbf{x}_t - \mathbf{x}^*)\|^2}{2\alpha_{t-1}(1-\beta_{1,t-1})} - \frac{\|\hat{\mathbf{V}}_T^{1/4}(\mathbf{x}_{T+1} - \mathbf{x}^*)\|^2}{2\alpha_T(1-\beta_{1,T})}$$

$$= \sum_{t=2}^T \left[\frac{\|\hat{\mathbf{V}}_t^{1/4}(\mathbf{x}_t - \mathbf{x}^*)\|^2}{2\alpha_t(1-\beta_{1,t})} - \frac{\|\hat{\mathbf{V}}_{t-1}^{1/4}(\mathbf{x}_t - \mathbf{x}^*)\|^2}{2\alpha_{t-1}(1-\beta_{1,t-1})}\right]$$

$$+ \frac{\|\hat{\mathbf{V}}_1^{1/4}(\mathbf{x}_1 - \mathbf{x}^*)\|^2}{2\alpha_1(1-\beta_{1,1})} - \frac{\|\hat{\mathbf{V}}_T^{1/4}(\mathbf{x}_{T+1} - \mathbf{x}^*)\|^2}{2\alpha_T(1-\beta_{1,T})}. \tag{88}$$

Further, the first term in RHS of (88) can be bounded as

$$\sum_{t=2}^{T}\left[\frac{\|\hat{\mathbf{V}}_t^{1/4}(\mathbf{x}_t-\mathbf{x}^*)\|^2}{2\alpha_t(1-\beta_{1,t})}-\frac{\|\hat{\mathbf{V}}_{t-1}^{1/4}(\mathbf{x}_t-\mathbf{x}^*)\|^2}{2\alpha_{t-1}(1-\beta_{1,t-1})}\right]$$

$$=\sum_{t=2}^{T}\left[\frac{\|\hat{\mathbf{V}}_t^{1/4}(\mathbf{x}_t-\mathbf{x}^*)\|^2}{2\alpha_t(1-\beta_{1,t})}-\frac{\|\hat{\mathbf{V}}_{t-1}^{1/4}(\mathbf{x}_t-\mathbf{x}^*)\|^2}{2\alpha_{t-1}(1-\beta_{1,t})}\right]$$

$$+\sum_{t=2}^{T}\left[\left(\frac{1}{1-\beta_{1,t}}-\frac{1}{1-\beta_{1,t-1}}\right)\frac{\|\hat{\mathbf{V}}_{t-1}^{1/4}(\mathbf{x}_t-\mathbf{x}^*)\|^2}{2\alpha_{t-1}}\right]$$

$$\overset{(a)}{\leq}\frac{1}{2(1-\beta_1)}\sum_{t=2}^{T}\left[\sum_{i=1}^{d}\left(\frac{\hat{v}_{t,i}^{1/2}(x_{t,i}-x_i^*)^2}{\alpha_t}-\frac{\hat{v}_{t-1,i}^{1/2}(x_{t,i}-x_i^*)^2}{\alpha_{t-1}}\right)\right]$$

$$\overset{(b)}{\leq}\frac{D_\infty^2\sum_{i=1}^{d}\sum_{t=2}^{T}\left[\frac{\hat{v}_{t,i}^{1/2}}{\alpha_t}-\frac{\hat{v}_{t-1,i}^{1/2}}{\alpha_{t-1}}\right]}{2(1-\beta_1)}\leq\frac{D_\infty^2\sum_{i=1}^{d}\hat{v}_{T,i}^{1/2}}{2\alpha_T(1-\beta_1)}\tag{89}$$

where the inequality (a) holds since $\beta_{1,t}\leq\beta_{1,t-1}\leq\beta_1$ and $1/(1-\beta_{1,t})-1/(1-\beta_{1,t-1})\leq0$, and the inequality (b) holds due to $\|x_t-x^*\|_\infty\leq D_\infty$ and $\frac{\hat{v}_{t,i}^{1/2}}{\alpha_t}-\frac{\hat{v}_{t-1,i}^{1/2}}{\alpha_{t-1}}\geq0$. Substituting (89) into (88), we obtain that

$$C\leq\frac{D_\infty^2\sum_{i=1}^{d}\hat{v}_{T,i}^{1/2}}{2\alpha_T(1-\beta_1)}+\frac{D_\infty^2\sum_{i=1}^{d}\hat{v}_{1,i}^{1/2}}{2\alpha_1(1-\beta_1)}\leq\frac{D_\infty^2\sum_{i=1}^{d}\hat{v}_{T,i}^{1/2}}{\alpha_T(1-\beta_1)},\tag{90}$$

where the last inequality holds since $\hat{v}_{t+1,i}^{1/2}\geq\hat{v}_{t,i}^{1/2}$ and $\alpha_1\geq\alpha_T$.

*We highlight that although the proof on bounding $C$ in [18, Theorem 4] is problematic, the conclusion of [18, Theorem 4] keeps correct.*

Substituting $C$ and $D$ into (86), we obtain that

$$\mathbb{E}\left[\sum_{t=1}^{T}\langle\hat{\mathbf{g}}_t,\mathbf{x}_t-\mathbf{x}^*\rangle\right]\leq\frac{\alpha\sqrt{1+\log T}\sum_{i=1}^{d}\mathbb{E}\|\hat{\mathbf{g}}_{1:T,i}\|}{(1-\beta_1)^2(1-\gamma)\sqrt{1-\beta_2}}$$

$$+\frac{D_\infty^2\sum_{i=1}^{d}\mathbb{E}[\hat{v}_{T,i}^{1/2}]}{\alpha_T(1-\beta_1)}+\frac{D_\infty^2}{2(1-\beta_1)}\sum_{t=1}^{T}\sum_{i=1}^{d}\frac{\beta_{1,t}\mathbb{E}[\hat{v}_{t,i}^{1/2}]}{\alpha_t}.\tag{91}$$

In (91), since $\sqrt{\cdot}$ is a concave function, the Jensen's inequality yields

$$\mathbb{E}[\sqrt{\hat{v}_{t,i}}]\leq\sqrt{\mathbb{E}[\hat{v}_{t,i}]}.\tag{92}$$

Substituting (92) into (91) and (78), we complete the proof.

□

# 4 Supplementary Material of Experiments

## 4.1 Problem and experiment setup

It is known that DNN-based image classifiers are vulnerable to adversarial examples—one can carefully craft images with imperceptible perturbations (a.k.a. adversarial perturbations or adversarial attacks) that can fool image classifiers even under a *black box* threat model, where details of the model are unknown to the attacker [5, 6, 48, 49].

We focus on two problem settings of black-box adversarial attacks: per-image adversarial perturbation and universal adversarial perturbation. Let $(\mathbf{x},t)$ denote a legitimate image $\mathbf{x}$ with the true label $t\in\{1,2,\ldots,K\}$, where $K$ is the total number of image classes. And let $\mathbf{x}'=\mathbf{x}+\boldsymbol{\delta}$ denote an adversarial example, where $\boldsymbol{\delta}$ is the adversarial perturbation. Our goal is to design $\boldsymbol{\delta}$ for a single

image $\mathbf{x}$ or multiple images $\{\mathbf{x}_i\}_{i=1}^{M}$. Spurred by [51], we consider the optimization problem

$$\begin{aligned}
\underset{\boldsymbol{\delta}}{\text{minimize}} \quad & \frac{\lambda}{M}\sum_{i=1}^{M} f(\mathbf{x}_i + \boldsymbol{\delta}) + \|\boldsymbol{\delta}\|_2^2 \\
\text{subject to} \quad & (\mathbf{x}_i + \boldsymbol{\delta}) \in [-0.5, 0.5]^d, \forall i,
\end{aligned} \tag{93}$$

where $f(\mathbf{x}_0 + \boldsymbol{\delta})$ denotes the (black-box) attack loss function, $\lambda > 0$ is a regularization parameter that strikes a balance between minimizing the attack loss and the $\ell_2$ distortion, and we normalize the pixel values to $[-0.5, 0.5]^d$. In problem (93), we specify the loss function for untargeted attack [51], $f(\mathbf{x}') = \max\{Z(\mathbf{x}')_t - \max_{j \neq t} Z(\mathbf{x}')_j, -\kappa\}$, where $Z(\mathbf{x}')_k$ denotes the prediction score of class $k$ given the input $\mathbf{x}'$, and the parameter $\kappa > 0$ governs the gap between the confidence of the predicted label and the true label $t$. In experiments, we choose $\kappa = 0$, and the attack loss $f$ reaches the minimum value 0 as the perturbation succeeds to fool the neural network.

In problem (93), if $M = 1$, then it becomes our first task to find per-image adversarial perturbations. If $M > 1$, then the problem corresponds to the task of finding universarial adversarial perturbations to $M$ images. Problem (93) yields a constrained formulation for the design of black-box adversarial attacks. Since some ZO algorithms are designed only for unconstrained optimization (see Table 1), we also consider the unconstrained version of problem (93) [24],

$$\begin{aligned}
\underset{\mathbf{w} \in \mathbb{R}^d}{\text{minimize}} \quad & \frac{\lambda}{M}\sum_{i=1}^{M}\big[f\big(0.5\tanh(\tanh^{-1}(2\mathbf{x}_i) + \mathbf{w})\big) \\
& +\|0.5\tanh(\tanh^{-1}(2\mathbf{x}_i) + \mathbf{w}) - \mathbf{x}_i\|_2^2\big],
\end{aligned} \tag{94}$$

where $\mathbf{w} \in \mathbb{R}^d$ are optimization variables, and we eliminate the inequality constraint in (93) by leveraging $0.5\tanh(\tanh^{-1}(2\mathbf{x}_i) + \mathbf{w}) = \mathbf{x}_i + \boldsymbol{\delta} \in [-0.5, 0.5]^d$.

The experiments of generating black-box adversarial examples will be performed on Inception V3 [45] under the dataset ImageNet [46]. We will compare the proposed ZO-AdaMM method with 6 existing ZO algorithms, ZO-SGD [9], ZO-SCD [22] and ZO-signSGD [14] for unconstrained optimization, and ZO-PSGD [27], ZO-SMD [23] and ZO-NES [6] for constrained optimization. The first 5 methods have been summarized in Table 1, and ZO-NES refers to the black-box attack generation method in [6], which applies a projected version of ZO-signSGD using natural evolution strategy (NES) based random gradient estimator. In all the aforementioned ZO algorithms, we adopt the random gradient estimator (14) and set $b = 1$ and $q = 10$ so that every method takes the same query cost per iteration. Accordingly, the total query complexity is consistent with the number of iterations.

In Fig. A1, we show the influence of exponential averaging parameters $\beta_1$ and $\beta_2$ on the convergence of ZO-AdaMM, in terms of the converged total loss while designing the per-image (ID 11 in ImageNet) and universal adversarial attack. As we can see, the typical choice of $\beta_2 > 0.9$ is no longer the empirically optimal choice in the ZO setting. In all of our experiments, we find that the choice of $\beta_1 \geq 0.9$ and $\beta_2 \in [0.3, 0.5]$ performs well in practice. In Table A1 and A2, we present the best learning rate parameter $\alpha$ founded by greedy search at each experiment, in the sense that the smallest objective function (corresponding to the successful attack) is achieved given the maximum number of iterations $T$.

**Figure A1:** The heat map of the converged objective value at 1000 iterations versus different combinations of $\beta_1$ and $\beta_2$ of ZO-AdaMM. (a) Unconstrained per-image (ID 11) adversarial attack problem (94); (b) Constrained per-image (ID 11) adversarial attack problem (93); (c) Universal adversarial attack problem (93) with $M = 10$.

| Methods | Learning rate $\alpha$ | Converged objective value | Success of attack |
|---|---|---|---|
| ZO-PSGD | $4 \times 10^{-4}$ | 245.92 | $\times$ |
| | $2 \times 10^{-4}$ | 78.66 | $\checkmark$ |
| | $1 \times 10^{-4}$ | 31.42 | $\checkmark$ |
| | $9 \times 10^{-5}$ | 30.98 | $\times$ |
| ZO-SMD | $8 \times 10^{-4}$ | 245.92 | $\times$ |
| | $5 \times 10^{-4}$ | 97.42 | $\checkmark$ |
| | $3 \times 10^{-4}$ | 35.19 | $\checkmark$ |
| | $9 \times 10^{-5}$ | 36 | $\times$ |
| ZO-NES | $5 \times 10^{-2}$ | 3997 | $\times$ |
| | $1 \times 10^{-2}$ | 194.22 | $\checkmark$ |
| | $9 \times 10^{-3}$ | 158.02 | $\checkmark$ |
| | $8 \times 10^{-3}$ | 129.30 | $\times$ |
| ZO-SCD | $8 \times 10^{-3}$ | 330.12 | $\times$ |
| | $2 \times 10^{-3}$ | 77.14 | $\checkmark$ |
| | $1 \times 10^{-3}$ | 42.87 | $\checkmark$ |
| | $9 \times 10^{-3}$ | 39.60 | $\times$ |
| ZO-SGD | $5 \times 10^{-3}$ | 1089.57 | $\times$ |
| | $8 \times 10^{-4}$ | 33.60 | $\checkmark$ |
| | $5 \times 10^{-4}$ | 31.11 | $\checkmark$ |
| | $4 \times 10^{-4}$ | 33.13 | $\times$ |
| ZO-signSGD | $8 \times 10^{-2}$ | 1590.02 | $\times$ |
| | $2 \times 10^{-2}$ | 113.43 | $\checkmark$ |
| | $1 \times 10^{-2}$ | 41.96 | $\checkmark$ |
| | $9 \times 10^{-3}$ | 39.60 | $\times$ |

**Table A1:** Greedy search on the best learning rate parameter $\alpha$ for generating per-image adversarial perturbations.

| Methods | Learning rate $\alpha$ | Converged objective value | Success of attack |
|---|---|---|---|
| ZO-PSGD | $1 \times 10^{-2}$ | 1072.05 | $\times$ |
| | $1 \times 10^{-3}$ | 147.46 | $\checkmark$ |
| | $4 \times 10^{-4}$ | 56.99 | $\checkmark$ |
| | $3 \times 10^{-4}$ | 36.86 | $\checkmark$ |
| | $2 \times 10^{-5}$ | 24.91 | $\times$ |
| ZO-SMD | $1 \times 10^{-2}$ | 788.46 | $\times$ |
| | $1 \times 10^{-3}$ | 60.98 | $\checkmark$ |
| | $6 \times 10^{-4}$ | 36.86 | $\checkmark$ |
| | $5 \times 10^{-4}$ | 29.56 | $\checkmark$ |
| | $4 \times 10^{-4}$ | 24.91 | $\times$ |
| ZO-NES | $1 \times 10^{-2}$ | 1230.15 | $\times$ |
| | $4 \times 10^{-2}$ | 107.74 | $\checkmark$ |
| | $7 \times 10^{-3}$ | 65.64 | $\checkmark$ |
| | $6 \times 10^{-3}$ | 54.00 | $\checkmark$ |
| | $5 \times 10^{-3}$ | 42.57 | $\times$ |

**Table A2:** Greedy search on the best learning rate parameter $\alpha$ for design of universal adversarial perturbations by solving problem (93).

## 4.2 Per-image black-box adversarial attack

We consider the task of per-image adversarial perturbation by solving problems (93) and (94), where $M = 1$ and $\lambda = 10$. In ZO-AdaMM (Algorithm 1), we set $\mathbf{v}_0 = \hat{\mathbf{v}}_0 = 10^{-5}$, $\mathbf{m}_0 = \mathbf{0}$, $\beta_{1t} = \beta_1 = 0.9$, $\beta_2 = 0.3$ and $T = 1000$. Here the exponential moving average parameters $(\beta_1, \beta_2)$ are exhaustively searched over $\{01, 0.3, 0.5, 0.7, 0.9\}^2$; see Fig. A1-(a) & (b) in Appendix 4 as an example. In ZO-AdaMM, we also choose a decaying learning rate $\alpha_t = \alpha/\sqrt{t}$ with $\alpha = 0.01$. For fair comparison, we use the decaying strategy for all other ZO algorithms, and we determine the best choice of $\alpha$ by greedy search over the interval $[10^{-4}, 10^{-2}]$; see Table A1 in Appendix 4 for more results on selecting $\alpha$.

In Table A3, we summarize the key statistics of each ZO optimization method for solving the per-image adversarial attack problem over 100 images randomly selected from ImageNet. For solving the unconstrained problem (94), ZO-SCD has the worst attack performance in general, i.e., leading to the largest number of iterations to reach the first successful attack and the largest final distortion.

We also observe that ZO-signSGD and ZO-AdaMM achieve better attack performance. However, the downside of ZO-signSGD is its poor convergence accuracy, given by the increase in distortion from the first successful attack to the final attack (i.e., $23.00 \rightarrow 28.52$ in Table A3). For solving the constrained problem (93), ZO-AdaMM achieves the best attack performance except for a slight drop in the attack success rate (ASR). Similar to ZO-signSGD, ZO-NES has a poor convergence accuracy in terms of the increase in $\ell_2$ distortion after the attack becomes successful.

| Problem | Methods | ASR | Ave. iters (1st succ.) | $\|\boldsymbol{\delta}_t\|_2^2$ (1st succ.) | Final $\|\boldsymbol{\delta}_T\|_2^2$ |
|---|---|---|---|---|---|
| (94) | ZO-SCD | 78% | 240 | 57.88 | 57.51 |
|  | ZO-SGD | 78% | **159** | 38.36 | 37.85 |
|  | ZO-signSGD | 74% | 179 | **23.00** | 28.52 |
|  | ZO-AdaMM | **81**% | 173 | 28.58 | **28.20** |
| (93) | ZO-NES | **82**% | 229 | 82.78 | 84.41 |
|  | ZO-PSGD | 78% | **112** | 60.32 | 58.10 |
|  | ZO-SMD | 76% | 198 | 35.08 | 35.05 |
|  | ZO-AdaMM | 78% | 197 | **23.77** | **23.72** |

**Table A3:** Performance of per-image attack over 100 images under $T = 1000$ iterations, where ASR represents attack success rate, and the distortion $\|\boldsymbol{\delta}\|_2^2$ is averaged over successful attacks only.

### 4.3 Universal black-box adversarial attack

In this experiment, we solve the constrained problem (93) for designing a universal adversarial perturbation $\boldsymbol{\delta}$, where we attack $M = 10$ images with the true class label 'brambling' and we set $\lambda = 10$ in (93). The setting of algorithmic parameters is similar to Appendix 4.2 except $T = 20000$. For ZO-AdaMM, we choose $\alpha = 0.002$, $\beta_1 = 0.9$, and $\beta_2 = 0.3$, where the sensitivity of exponential moving average parameters $(\beta_1, \beta_2)$ is shown in Fig. A1-(c). For the other ZO algorithms, we greedily search $\alpha$ over $[10^{-2}, 10^{-4}]$ and choose the value that achieves the best convergence accuracy as shown in Table A2.

In Fig. A2, we visualize the pattern of universal adversarial perturbation obtained from different methods. As we can see, the resulting universal perturbation pattern identifies the most discriminative image regions corresponding to the true label 'brambling'. We also observe that although each method successfully generates the black-box adversarial example, ZO-AdaMM yields the strongest attack that requires the least distortion strength.

**Figure A2:** Visualization of universal perturbation versus different iterations and the eventually generated adversarial examples. Left four columns present universal perturbations found by different ZO algorithms at the iteration number 1000, 5000, 10000 and 20000, where the depth of the color corresponds to the strength of the perturbation, and the maximum distortion $\|\boldsymbol{\delta}\|_\infty$ (with deepest green) is given at the bottom of each subplot. The right four columns are 4 of 10 adversarial examples that lead to missclassfication from the original label 'brambling' to an incorrectly predicted label given at the bottom of each subplot.