[Reviews · NeurIPS 2019]

Reviewer 1



The paper proposed a novel zeroth-order optimization algorithm inspired by the success of adaptive gradient methods (such as ADAM, NADAM, AMSGRAD, etc) in first-order optimzation. It is the first work leveraging this into block-box optimization which is a very important topic in machine learning and I believe this work has potential for a high impact. The theoretical analysis shows a convergence rate which is poly(d) worse than first-order AdaMM methods, which is acceptable for zeroth-order algorithms. The paper is well and clearly written.

Reviewer 2



The context of this work is derivative-free optimization (DFO). The proposed method is an adaptation of the AdaMM first-order algorithm. However, the DFO literature has been ignored. Many existing algorithms may be applied to solve the blackbox optimization problem. It is not clear to what the proposed method is compared to. It seems that all competitors fall into the same category and may have been recoded by the authors. However many DFO solvers are available. In these conditions, it is not possible to assess the quality of the proposed algorithm. UPDATE: The authors responded very positively to my comments. They are eager to include au augmented literature review on DFO methods.

Reviewer 3



This paper proposes a zeroth-order adaptive momentum method for black-box optimization, by approximating the stochastic gradient using the forward difference of two function values at a random unit direction. The paper also shows the convergence analysis in terms of Mahalanobis distance for both unconstrained and constrained nonconvex optimization with the ZO-AdaMM, which results in sublinear convergence rates that are roughly a factor of the square root of dimension worse than that of the first-order ZO-AdaMM, as well as for constrained convex optimization. The proposed scheme is quite interesting, which solves the (non)convex optimization in a new perspective, and somewhat provides new insight to the adaptive momentum methods. In particular, the paper provides a formal conclusion that the Euclidean projection may results in non-convergence issue in stochastic optimization. The paper also shows the applications to black-box adversarial attacks problems and validate the method by comparing it with other ZO methods. Overall, the paper is well written and easy to follow, including the proof I have checked. Although this work is technically sound, the authors may consider the following points to improve the quality and significance on top of the current presence. 1. As the first-order adaptive momentum methods are still quite popular and in practice, gradients are not quite difficult to be obtained, what is the practical advantage of ZO-AdaMM compared with the first-order adaptive momentum method? Although the authors provide a table to show the comparison clearly, unfortunately, in the experiments, no first-order methods are used for comparison. 2. I believe the ZO-AdaMM is not only proposed for adversarial learning, like the experiments shown in the paper. Why do the authors not show any experiments for regular image classification to check the performance of the proposed approach? Or ZO-AdaMM is better for the applications to black-box adversarial attacks? If the ZO-AdaMM was proposed as a generic optimizer for most of deep learning problems, I think it should be good to see more applications to different datasets. 3. In the paper, the authors say the tolerance on the smoothing parameter \mu is an important factor to indicate the convergence performance of ZO optimization methods. I just wonder in the paper how the authors would select \mu such that a good gradient approximation can be made. 4. In the Eq. (65), the definition A has an equal sign, what does it mean in the proof? 5. After Proposition 1, the authors introduced Mahalanobis distance, it would be better if the authors could give more motivation and intuition on this metrics and why this is a good choice for ZO-AdaMM, other than just showing the limitedness of the Euclidean projection 6. In the proof of Proposition 1, when the authors show the optimality condition of (26), does it mean an inequality? The same to (27). 7. In the supplementary materials, in Section 4.2, the authors set the choice of step size by greedy search over the interval [10^{-2}, 10^{-4}] (This interval seems inverse?). However, in the table A1 and A2, a couple of learning rates are out of this interval. Why is that? Also, how to relate the converged objective values to successes of attack in the paper?

[Author Response · NeurIPS 2019]

**Additional Experiments (AE)**. Beyond the generation of per-image and universal adversarial attacks, we conduct an
additional experiment (in response to R#1 & R#3): *sensor selection via ZO optimization* introduced by [1]. The goal of
sensor selection is to seek the optimal trade-off between sensor activations and field estimation accuracy. The rationale
behind the use of ZO optimization (ZOO) is to avoid the complex gradient computation that requires matrix inversion.
In response to R#2, we compare ZO-AdaMM with a derivative-free optimization (DFO) solver `COBYLA`. Our results
show that ZO-AdaMM yields *6.4%* lower object value and saves *37.7%* computation time under the same query number
*500*. More detailed comparisons with DFO and ZOO methods will be added in the revision.

**Reviewer#1.** Thanks for the comments! We have added a new application on sensor selection illustrated in above **AE**.

**Reviewer#2.** [What the proposed method is compared to] $\rightarrow$ ZO-AdaMM is compared to the class of ZOO methods,
which utilize function difference based *random gradient estimates*; see Sec. 1 (lines 28-43) and Sec. 3 (lines 112-124). In
spite of DFO, the literature on ZO counterparts of first-order algorithms has been vast. Different from direct search (DS)
and model-based DFO methods [2,3], ZO-AdaMM is the first algorithm that bridges the random gradient estimation and
the adaptive gradient method, where the latter is quite popular in the current DL/ML applications. Both our convergence
(Table 1) and empirical results (comparison with 6 state-of-the-art ZOO methods) showed the quality of the proposed
algorithm. [DFO literature and comparison] $\rightarrow$ It is indeed valuable to enrich our related work on DFO. Thanks! In the
revision, we will review DS-based and model-based methods, and commonly-used DFO solvers [3]. There is also a
connection from the simplex gradient [2] (in the linear model based DFO) to the randomized gradient estimation (in
ZOO). We will compare our method with existing DFO solvers, e.g., `PSwarm` and `NOMAD` for DS methods and `COBYLA`
and `BOBYQA` for model-based methods. A preliminary comparison with `COBYLA` was illustrated in **AE**.

**Reviewer#3.** [ZO-AdaMM versus first-order AdaMM]$\rightarrow$ ZO-AdaMM belongs to the class of ZOO methods, and its
advantages appear when the gradient is *(a) impossible* or *(b) difficult* to obtain. For example, the design of adversarial
examples falls into the case (a). The sensor selection example introduced in **AE** belongs to the case (b). If the gradients
are known and easily computed, then ZO-AdaMM is not better than its first-order counterpart due to its worse dimension
dependency; see Table 1. Since our experiments focus on *black-box* adversarial attacks, the first-order method would
*not* be available in fact. However, following the comment, we perform the additional comparison between ZO-AdaMM
and AdaMM in generating per-image adversarial perturbation. Not surprisingly, AdaMM reaches a better solution in
terms of $43.6\%$ reduction in averaged $\ell_2$ perturbation and $11.8\%$ enhancement in averaged attack success rate over $100$
ImageNet images. [Adversarial learning & other applications]$\rightarrow$ The research in adversarial robustness of DL modes has
rapidly gaining its popularity and attention in the past two years, e.g., design of black-box attacks at Adversarial Vision
Challenge, NeurIPS'18. Many benchmark black-box attack methods were built on ZO optimization, e.g., ZO-SignSGD
and ZO-NES (compared in the paper). Thus, we focus on the application in adversarial learning. Notably, ZO-AdaMM
significantly outperforms 7 existing methods. However, we also conduct a new sensor selection experiment; see **AE**.
[Choice of $\mu$]$\rightarrow$ It is shown from Eq. (19) that $\mu$ controls the bias of the gradient estimate. To obtain the desired
sub-linear convergence rate, the existing work has to select $\mu$ small enough. However, this causes numerical issues
[4]: the stochastic function difference could be dominated by the stochastic noise and fails to represent the function
differential. Thus, the mildness of $\mu$ is an important metric. In the original experiments, we set $\mu = 5 \times 10^{-3}$ obeying
the order of $O(1/\sqrt{d})$ since $d \gg T$, where $d$ is dimension of ImageNet image, and $T$ is number of iterations. We also
conduct a more careful tuning on $\mu$ by searching 5 points in $[5 \times 10^{-4}, 5 \times 10^{-2}]$. We observe that $\mu = 2 \times 10^{-3}$
yields the best result (in terms of the converged loss value) but with only a minor improvement (3.7%) compared to our
original choice. [A in (65)]$\rightarrow$ $A$ refers to the sum of the first two terms at RHS of (65) (without the equal sign). [Why
using Mahalanobis (M-) distance]$\rightarrow$ M-distance facilitates our convergence analysis in an equivalently transformed
space, over which the analysis can be generalized from the conventional projected gradient descent framework. To
get intuition, let us consider a simpler first-order case with the **x**-descent step given by Algorithm 1 as $\beta_{1,t} = 0$ and
$\mathcal{X} = \mathbb{R}^d$: $\mathbf{x}_{t+1} = \mathbf{x}_t - \alpha \hat{\mathbf{V}}_t^{-1/2} \nabla f(\mathbf{x}_t)$. Note that the ZO case is more involved but follows the same intuition. Upon
defining $\mathbf{y}_t \triangleq \hat{\mathbf{V}}_t^{1/4} \mathbf{x}_t$, the **x**-update can then be rewritten as the update rule in **y**: $\mathbf{y}_{t+1} = \mathbf{y}_t - \alpha \hat{\mathbf{V}}_t^{-1/4} \nabla f(\mathbf{x}_t)$. Since
$\nabla_{\mathbf{y}_t} f(\mathbf{x}_t) = (\frac{\partial \mathbf{x}_t}{\partial \mathbf{y}_t})^T \nabla f(\mathbf{x}_t) = \hat{\mathbf{V}}_t^{-1/4} \nabla f(\mathbf{x}_t)$, the **y**-update, $\mathbf{y}_{t+1} = \mathbf{y}_t - \alpha \nabla_{\mathbf{y}} f(\mathbf{x}_t)$, obeys the gradient descent
framework. In the constrained case, a similar but more involved analysis can be made, showing that the *M-projection in*
*the* **x***-coordinate system* is *equivalent* to the *Euclidean projection in the* **y***-coordinate system* which makes projected
gradient descent applicable to the update in **y**. And the direct use of *Euclidean projection in the* **x***-coordinate system*
leads to *divergence* in ZO-AdaMM (Prop. 1). [Typos in (26) & (27)] $\rightarrow$ Yes, "$\geq 0$" should be added at the end of
equations. [Choice of stepsize] $\rightarrow$ Yes, the stepsize interval should be reversed. In Table A1-A2, a stepsize out of the
range was included to show that the attack becomes *unsuccessful* when the stepsize is below our choice (e.g., $9 \times 10^{-5}$
in Table A1 for ZO-PSGD), namely, the further reduction of stepsize does *not* improve the attack performance.

[1] Liu, et al., "ZO ADMM ....", AISTATS'18. [2] Audet & Hare, "Derivative-free and blackbox optimization", 2017. [3] Rios, et al.,
"DFO: a review ...", 2013. [4] Lian, et al., "A comprehensive linear speedup ..." NIPS'16.


[Meta-Review · NeurIPS 2019]

All the reviewers agreed that the paper is novel, interesting, and worth to be published in NeurIPS. Please take into account the reviewers' comment in preparing the camera-ready version, especially the expansion of the literature on DFO methods.